



# Airborne observations of the surface cloud radiative effect during different seasons over sea ice and open ocean in the Fram Strait

Sebastian Becker[1], André Ehrlich[1], Michael Schäfer[1], and Manfred Wendisch[1]

[1]Leipzig Institute for Meteorology (LIM), Leipzig University, Leipzig, Germany

**Correspondence:** Sebastian Becker (sebastian.becker@uni-leipzig.de)

**Abstract.** This study analyzes the surface cloud radiative effect (CRE) obtained during airborne observations of three campaigns in the Arctic north-west of Svalbard. The surface CRE quantifies the potential of clouds to modify the radiative energy budget of the surface and is calculated by combining broadband radiation measurements during low-level flight sections in mostly cloudy conditions with radiative transfer simulations of cloud-free conditions. The significance of surface albedo changes due to the presence of clouds is demonstrated and this effect is considered in the cloud-free simulations. The observations are discussed with respect to differences of the CRE between sea ice and open ocean surfaces, and between the seasonally different campaigns. The results indicate that the CRE depends on both cloud, illumination, surface, and thermodynamic properties. The solar and thermal-infrared (TIR) component of the CRE are analyzed separately and in combination. The inter-campaign differences of the solar CRE are dominated by the seasonal cycle of the solar zenith angle, with the largest cooling effect in summer. The lower surface albedo causes a larger solar cooling effect over open ocean than over sea ice, which amounts to $-259\,\mathrm{W\,m^{-2}}$ ($-108\,\mathrm{W\,m^{-2}}$) and $-65\,\mathrm{W\,m^{-2}}$ ($-17\,\mathrm{W\,m^{-2}}$), respectively, during summer (spring). Independent of campaign and surface type, the TIR CRE is only weakly variable and shows values around $75\,\mathrm{W\,m^{-2}}$. In total, clouds show a cooling effect over open ocean during all campaigns. In contrast, clouds over sea ice exert a warming effect to the surface, which neutralizes during mid-summer. Given the seasonal cycle of the sea ice distribution, these results imply that clouds in the Fram Strait region cool the surface during the sea ice minimum in late summer, while they warm the surface during the sea ice maximum in spring.

## 1 Introduction

The enhanced warming and the rapid loss of sea ice are the most obvious signs of accelerated climate changes currently ongoing in the Arctic. Because these changes appear much faster compared to the rest of the globe, the term Arctic amplification was introduced (Serreze and Barry, 2011; Wendisch et al., 2017, 2022a). A multitude of atmospheric processes and feedback mechanisms contributes to this amplified transformation of the Arctic climate system. Clouds play a substantial, yet not fully understood, role in Arctic amplification by their involvement in several feedbacks. For example, the downward radiative energy fluxes in the thermal-infrared (TIR) spectral range ($\sim 4\text{–}100\,\mu\mathrm{m}$) are increased by clouds, which leads to a warmer surface and delayed refreezing, thinner sea ice, and faster melting (positive cloud–sea ice feedback, Morrison et al., 2019). At the same time, the increased fraction of liquid water in the clouds increases the cloud optical thickness and the reflection of solar radiation ($\sim$





0.3–4 μm) by the cloud (negative cloud optical thickness feedback, e. g., Zelinka et al., 2012; Ceppi et al., 2015). Furthermore, the indirect influence of clouds on other feedbacks is demonstrated by the reduction of the sea ice–albedo feedback in summer through increased cloud fraction or optical thickness (Kay et al., 2012; Choi et al., 2020). This multitude of partly opposing effects complicates the evaluation of the combined impact of clouds within the Arctic climate system and makes the sign

(warming or cooling) of the total cloud feedback uncertain (Forster et al., 2021).

In consequence, a realistic representation of the impact of clouds within the Arctic climate system in numerical weather and climate models appears crucial. In particular, the radiative energy budget (REB) of the surface and the atmosphere is largely determined by the presence and properties of clouds (Wendisch et al., 2022b). The REB is quantified by the difference of downward and upward irradiances, $F^\downarrow$ and $F^\uparrow$, respectively; it is referred to as net irradiance $F_{\text{net}}$, with:

$$F_{\text{net}} = F^\downarrow - F^\uparrow. \tag{1}$$

The cloud impact on the REB is quantified by the cloud radiative effect (CRE), which is also referred to as cloud radiative forcing (Ramanathan et al., 1989). It is derived from the difference of the net irradiances in cloudy ($F_{\text{net,cld}}$) and cloud-free ($F_{\text{net,cf}}$) atmospheric conditions:

$$\Delta F = F_{\text{net,cld}} - F_{\text{net,cf}}. \tag{2}$$

The CRE depends on both, microphysical (e. g., cloud phase; liquid water path, LWP; and effective radius, $r_{\text{eff}}$) and macro-physical (e. g., cloud fraction; cloud height) cloud properties, but also on the thermodynamic circumstances, the solar zenith angle (SZA), and the surface albedo (Shupe and Intrieri, 2004).

In contrast to the cooling effect of clouds on global average (Allan, 2011), long-term, ground-based observations at single locations around the Arctic identified an average warming effect of clouds at the surface, ranging from $3.5\,\text{W}\,\text{m}^{-2}$ to $33\,\text{W}\,\text{m}^{-2}$

(Dong et al., 2010; Intrieri et al., 2002; Miller et al., 2015; Ebell et al., 2020). While the TIR warming effect dominates for the frequently occurring, relatively warm low-level clouds that are typically related to distinct temperature inversions, the solar cooling potential of the clouds is limited by the high surface albedo and SZA (during polar day) characteristic for the Arctic (e. g., Curry et al., 1996; Shupe and Intrieri, 2004). During summer, however, the relatively low SZA causes the solar cooling effect to dominate over the TIR warming effect (Ebell et al., 2020; Dong et al., 2010; Intrieri et al., 2002), which determines a

seasonal cycle of the CRE. Spatial differences of the environmental conditions among the measurement sites mainly cause the variability of the surface CRE found in the literature. For example, the average CRE observed during the Surface HEat Budget of the Arctic Ocean (SHEBA) drift experiment (Intrieri et al., 2002), or on the Greenland ice sheet (Summit, Miller et al., 2015), where the surface was covered by snow or ice all year round, is larger compared to the partly snow-free land surfaces at Barrow (Dong et al., 2010) or Ny-Ålesund (Ebell et al., 2020).

The comparison of the different CRE results is further hampered by the inconsistent consideration of the cloud impact on the thermodynamic profiles and on the surface albedo (Stapf et al., 2021a, 2020). Some studies applied the radiative-transfer approach, where the cloud-free state is simulated by only removing the cloud, neglecting adjustments of the thermodynamic profiles and the surface albedo between cloudy and cloud-free conditions (e. g., Intrieri et al., 2002). Others determined the





CRE from measurements only, which were obtained during cloudy and cloud-free conditions (e. g., Dong et al., 2010). This

measurement-based approach accounts for the adjustment effects (Stapf et al., 2020). The resulting differences between the two

approaches can be significant. Stapf et al. (2021a) demonstrated that, due to the decreased surface temperature, the TIR CRE

obtained during SHEBA would be up to $25\,\mathrm{W\,m^{-2}}$ lower in autumn if the measurement-based approach was used. In summer,

no significant differences were found. Since the temporal dependence of the thermodynamic adjustments to cloud dissipation

complicates an accurate and continuous quantification of this effect (Walsh and Chapman, 1998; Wendisch et al., 2022b),

differences between both approaches will remain and likely depend on cloud type, season, and surface conditions. In contrast,

the conceptual differences resulting from the cloud-induced surface albedo change can be reduced for the radiative-transfer

approach. For snow-covered surfaces, Stapf et al. (2020) applied an albedo parameterization to obtain the surface albedo in

cloud-free conditions and found a doubling of the solar cooling effect in the Fram Strait at the beginning of the melting season

when the retrieved albedo was used for the calculation of the cloud-free net irradiance.

The majority of previous CRE studies in the Arctic as well as the investigations of the cloud impact on the surface albedo

and the CRE (Stapf et al., 2020) were conducted over sea ice or mostly snow-covered surfaces. Less efforts have been spent to

study the CRE over open (ice-free) ocean (Kay et al., 2016), although this situation will become more dominant in the future

Arctic. Over an open ocean, quite different characteristics of the mean CRE and the cloud impact on the surface albedo appear.

However, measurements of the REB and the CRE are difficult to obtain over open ocean. Shipborne radiation measurements

mostly consist of the downward irradiances only and rely on assumptions or complementary (e. g., satellite) measurements of

surface albedo, near-surface air temperature, and surface emissivity to calculate the upward irradiances (Protat et al., 2017;

Barrientos-Velasco et al., 2022). Polavarapu (1978) obtained the net irradiance from a combination of two pairs of radiometers,

which were mounted at the left and right outsides of the ship's structure and shadowed at the half facing towards the ship. Kay

and L'Ecuyer (2013) used satellite observations to derive the surface CRE and found an annual mean CRE similar to that from

Ebell et al. (2020). However, both results included snow-covered and snow-free observations.

To characterize the CRE over the individual sea ice and open ocean surfaces in close proximity to each other , this study

uses airborne measurements of broadband downward and upward irradiances combined with radiative transfer simulations. For

this purpose, the data from three airborne campaigns performed during different seasons are analyzed. Section 2 introduces the

measurements as well as the campaigns and their surface and meteorological characteristics. The radiative transfer simulations

and the effect of the cloud-induced albedo change over open ocean are described in Sect. 3. The solar and the TIR CRE as well

as their variability as a function of the surface type, and between the different campaigns are assessed in Sect. 4. Conclusions

are given in Sect. 5

## 2   Observations

### 2.1   Airborne campaigns and instrumentation

To study atmospheric processes in the lower Arctic troposphere, airborne measurements of cloud, surface, and thermodynamic

properties were performed during three seasonally distinct campaigns in the vicinity of Svalbard. The Airborne measurements





**Table 1.** Statistical overview of the analyzed campaigns.

| Campaign | AFLUX | ACLOUD | MOSAiC-ACA |
|---|---|---|---|
| Period | 19 March–11 April 2019 | 23 May–26 June 2017 | 30 August–13 September 2020 |
| Aircraft | *Polar 5* | *Polar 5, Polar 6* | *Polar 5* |
| Total flight hours | 67.5 | 178.5 | 44.3 |
| Low-level flight hours | 6.1 | 13.6 | 1.5 |
| Observations over sea ice (%) | 65 | 50 | 2 |
| Observations over open ocean (%) | 16 | 15 | 72 |
| Observations over the MIZ (%) | 19 | 35 | 26 |
| Median SZA (°) | 76 | 60 | 76 |

of radiative and turbulent FLUXes of energy and momentum in the Arctic boundary layer (AFLUX) campaign was conducted in early spring 2019 (Mech et al., 2022), while the Arctic CLoud Observations Using airborne measurements during polar Day (ACLOUD) campaign was performed in late spring/early summer 2017 (Wendisch et al., 2019). Additionally, the Multidisci-

plinary drifting Observatory for the Study of Arctic Climate – Airborne observations in the Central Arctic (MOSAiC-ACA) campaign was conducted in late summer 2020 and accompanied the MOSAiC drift expedition with airborne measurements (Shupe et al., 2022). Table 1 lists the periods during which the campaigns were performed. During ACLOUD, the measurements were accomplished onboard the two research aircraft *Polar 5* and *Polar 6* from Alfred Wegener Institute, Helmholtz Centre for Polar and Marine Research (Wesche et al., 2016). During the other campaigns, only *Polar 5* was operated. The

majority of the observations took place in the eastern Fram Strait north-west of Svalbard, the corresponding flight tracks are displayed in Fig. 1. Depending on the campaign, between 5 % and 20 % of the total flight time was dedicated to low-level flight sections, not exceeding a flight altitude of 250 m.

During the low-level sections, the broadband upward and downward irradiances were measured with a pair of downward- and upward-directed pyranometers (sensitive in the solar spectral range between 0.2–3.6 μm) and pyrgeometers (sensitive in

the TIR range between 4.5–42 μm) at a frequency of 20 Hz. From these measurements, the net irradiance (Eq. 1) and the surface albedo (the ratio of solar upward and downward irradiances) in mostly cloudy conditions were derived. For aircraft attitudes exceeding 5° in roll and pitch angle, the irradiance data that are related to a horizontal reference plane, had to be discarded. This, and the exclusion of data due to severe icing of the instruments, led to a reduction of the low-level data set by 41 %, 63 %, and 44 % for AFLUX, ACLOUD, and MOSAiC-ACA, respectively. The remaining low-level flight time

used for the analysis is given in Table 1. Additionally, the surface brightness temperature and the sea ice concentration were derived from measurements of a nadir-directed Kelvin infrared radiation thermometer (KT-19, sampling frequency of 20 Hz), and a three-channel digital camera equipped with a 180° fish-eye lens (sampling frequency of 1/6 Hz). The individual pixels of the radiance-calibrated images of the fish-eye camera were classified into the different surface types based on their reflection





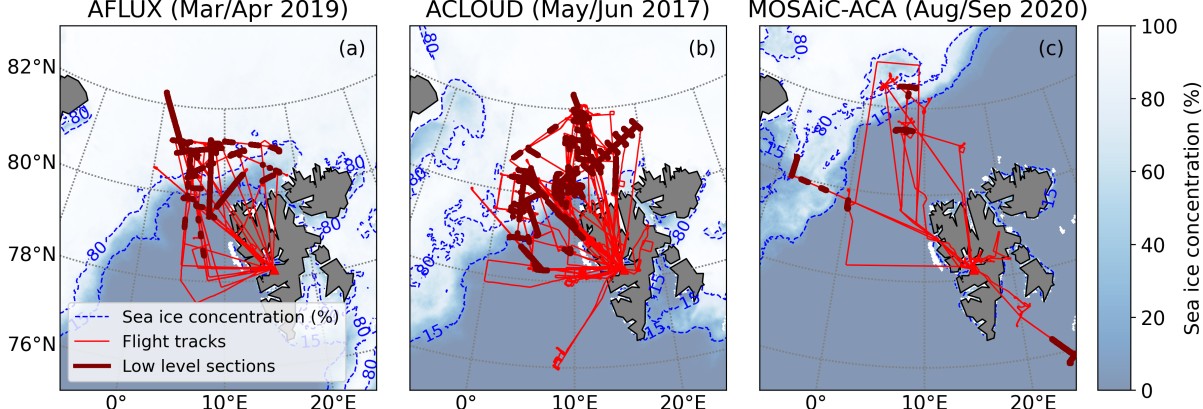

**Figure 1.** Flight tracks (red) and low-level sections (bold brown) performed during (a) AFLUX, (b) ACLOUD, and (c) MOSAiC-ACA. Each panel shows the mean sea ice concentration (derived from space-borne observations, Spreen et al., 2008) present during the respective campaign, the dashed blue lines indicate the 15 % and 80 % isolines of sea ice concentration that confine the MIZ according to the definition of Strong and Rigor (2013).

characteristics (Becker et al., 2022), in order to derive the cosine-weighted surface type fraction of each image. Based on the transmissivity of the clouds, which is the fraction of the downward irradiances measured in mostly cloudy atmospheric conditions and simulated for cloud-free conditions, an equivalent LWP was retrieved using the method of Stapf et al. (2020) and assuming clouds with a droplet $r_{eff}$ of 8 μm. Despite the neglected cloud ice and the fixed $r_{eff}$ in the retrieval, the equivalent LWP provides a robust estimate of the optical thickness of the clouds. Regular profiles of temperature and relative humidity were obtained using the in situ meteorological measurements during aircraft ascents and descents, sondes dropped from the aircraft, and, for the higher atmosphere, radiosoundings launched at Ny-Ålesund. Measurements of profiles of the cloud liquid water content (LWC) obtained from various in situ cloud probes were used to retrieve the cloud boundaries. Further details on the aircraft instrumentation and data processing are provided by Ehrlich et al. (2019b) and Mech et al. (2022).

## 2.2 Sea ice situation during the campaigns

The different distributions of sea ice in the Fram Strait region during the three campaigns investigated in this paper are depicted in Fig. 1 based on the space-borne observations of the sea ice concentration (Spreen et al., 2008). While the sea ice edge was roughly located between 80–81° N during AFLUX (Fig. 1a), it was situated slightly further south during ACLOUD (Fig. 1b). During both campaigns, the region east of Svalbard was covered by almost closed sea ice. Since MOSAiC-ACA was performed temporally close to the annual sea ice minimum, ice-free conditions were present east of the island and the sea ice edge in the Fram Strait was mostly north of 82° N. It was reached by *Polar 5* only for a short flight section (Fig. 1c). Consequently, more than half of the low-level observations during AFLUX and ACLOUD were performed over closed sea ice, while during MOSAiC-ACA, the fraction of observations over sea ice amounts to only 2 % and strongly limits the statistical representation of this situation. Instead, almost three quarters of the low-level observations were performed over open ocean during MOSAiC-





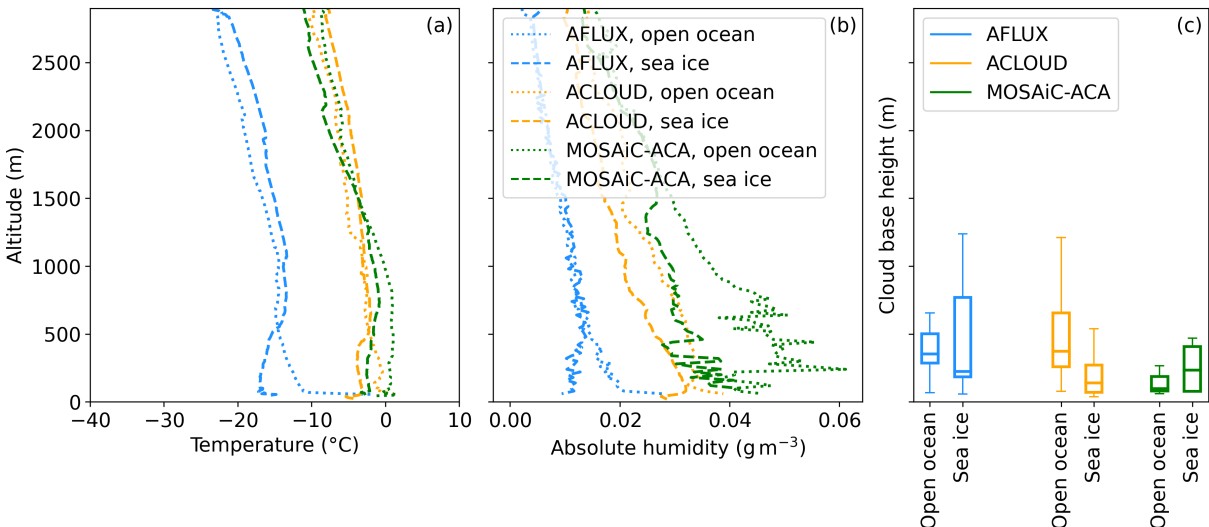

**Figure 2.** Mean profiles of (a) temperature, and (b) absolute humidity obtained during aircraft ascents and descents of the individual campaigns (colour-coded), separated for the different surface types (linestyle-coded). (c) Box-whisker plots of the cloud base height obtained from the in situ cloud probes (see text for details). The surface type separation is based on the space-borne observations of the sea ice concentration (Spreen et al., 2008) and the surface type definition of Strong and Rigor (2013).

ACA compared to about 15 % during the other campaigns (Table 1). Note that for the low-level observations, the MIZ comprises all observations with $f_{ice}$ between 0.05 and 0.95 and, thus, deviates from the MIZ definition of Strong and Rigor (2013), where

$f_{ice}$ is between 0.15 and 0.8. These modified thresholds are motivated by the strong impact of the surface type on the surface radiative properties, which significantly changes already for small fractions of sea ice or open ocean and requires a more rigorous separation of the MIZ (e. g., Becker et al., 2022).

### 2.3    Thermodynamic profiles and cloud base height

Mean profiles of various thermodynamic properties measured during the campaigns are shown in Figs. 2a and 2b. According

to the time of the year, AFLUX showed significantly colder temperatures compared to the remaining campaigns (Fig. 2a). While the mean near-surface temperature over sea ice was $-16\,°C$ during AFLUX, it was around $-3\,°C$ during ACLOUD and MOSAiC-ACA. Although the temperature at the open ocean surface was close to the freezing point during all campaigns, it strongly decreased within the lowermost $80\,m$ during AFLUX. Over sea ice, a surface-based temperature inversion was present during all campaigns, which was strongest during AFLUX. In contrast, temperature inversions were weak or absent over open

ocean, with the least stable mean profile obtained for AFLUX. In higher altitudes, the mean profiles over sea ice and open ocean of the same campaign agreed well.

    The emission and absorption by the atmospheric water vapour relevant for the surface TIR irradiances, depends on the absolute humidity (AH), which is shown in Fig. 2b. Due to the lower equilibrium pressure and the resulting lower concentration





of water vapour for colder temperatures, AFLUX showed the lowest AH. The differently shaped temperature profiles over sea
ice and open ocean below 500 m are imprinted in the mean AH profile. Despite the comparable temperature range, the AH was
significantly lower during ACLOUD than during MOSAiC-ACA, which is due to the lower relative humidity (RH, not shown).
Furthermore, the lower RH over sea ice reduced the AH compared to open ocean during ACLOUD. During MOSAiC-ACA,
both the higher temperature and the larger RH caused the increased AH over open ocean below 1500 m.

The surface TIR CRE largely depends on the cloud base temperature, which is determined by the location of the cloud within
the temperature profile. Figure 2c shows statistics of the cloud base height obtained from the in situ cloud probes. Clouds were
identified by a LWC threshold of $0.005\,\mathrm{g\,cm^{-3}}$. During AFLUX, the median cloud base height over open ocean was 352 m.
Over sea ice, the median cloud base height was lower (223 m) but more variable. The typical cloud base temperature range is
estimated from the mean temperature (Fig. 2a) in the altitudes corresponding to the inter-quartile range (IQR) of the cloud base
height. For AFLUX, the cloud base temperature was in the range of $-14\,^{\circ}\mathrm{C}$ over sea ice, and between $-13\,^{\circ}\mathrm{C}$ and $-17\,^{\circ}\mathrm{C}$
over open ocean. During ACLOUD, the median cloud base height of 372 m over open ocean was similar compared to AFLUX,
while clouds over sea ice were significantly lower (140 m). The resulting cloud base temperatures range around $-3$–$0\,^{\circ}\mathrm{C}$ and
$-3\,^{\circ}\mathrm{C}$ over open ocean and sea ice, respectively. During MOSAiC-ACA, the median cloud base height over sea ice was 234 m,
while clouds over open ocean showed very low bases (97 m). The cloud base temperatures were about $0\,^{\circ}\mathrm{C}$ and $-2\,^{\circ}\mathrm{C}$ over
open ocean and sea ice, respectively. In general, the slightly colder cloud base temperatures over sea ice seem to result rather
from the colder low-level temperatures than from the different cloud base heights.

## 2.4   Cloud liquid water path

The statistical characteristics of the equivalent cloud LWP assembled during the campaigns are illustrated in Fig. 3 as a rough
measure for optical thickness. During AFLUX (Fig. 3a), clouds with an equivalent LWP below $10\,\mathrm{g\,m^{-2}}$ were most frequent
over sea ice, while the largest mode of equivalent LWP over open ocean occurs between $30\,\mathrm{g\,m^{-2}}$ and $50\,\mathrm{g\,m^{-2}}$. The TIR CRE
is especially sensitive to the LWP of optically thin clouds below $30\,\mathrm{g\,m^{-2}}$ (e. g., Shupe and Intrieri, 2004; Ebell et al., 2020),
but almost constant for larger LWPs. Accordingly, this threshold was used to distinguish between thin and thick clouds, which
are analyzed separately. Corresponding to the PDFs, the meidan equivalent LWP of thin clouds is lower over sea ice than over
open ocean, with values of $15\,\mathrm{g\,m^{-2}}$ and $18\,\mathrm{g\,m^{-2}}$, respectively (Fig. 3d). In contrast, the median of thick clouds is larger over
sea ice. However, thin clouds occurred more frequently over sea ice compared to open ocean (numbers in Fig. 3a).

The PDF of the equivalent LWP derived from ACLOUD measurements (Fig. 3b) reveals a similar distribution of thin clouds
over sea ice and open ocean. Both surface types showed similar median values for thin clouds (around $17\,\mathrm{g\,m^{-2}}$, Fig. 3e). Over
open ocean, clouds with an equivalent LWP of $30$–$50\,\mathrm{g\,m^{-2}}$ were most common, while larger LWPs were more frequent over
sea ice than over open ocean. Thus, the median equivalent LWP of thick clouds was larger over sea ice. The thin cloud fraction
was slightly lower over sea ice compared to open ocean. Compared to AFLUX, thin clouds occurred significantly less frequent
over both surface types.

During MOSAiC-ACA (Fig. 3c), the vast majority of the clouds over the sparsely sampled sea ice showed an extremely
low equivalent LWP. The median equivalent LWP of thin clouds was $7\,\mathrm{g\,m^{-2}}$ (Fig. 3f) and almost 90 % of the sampled clouds





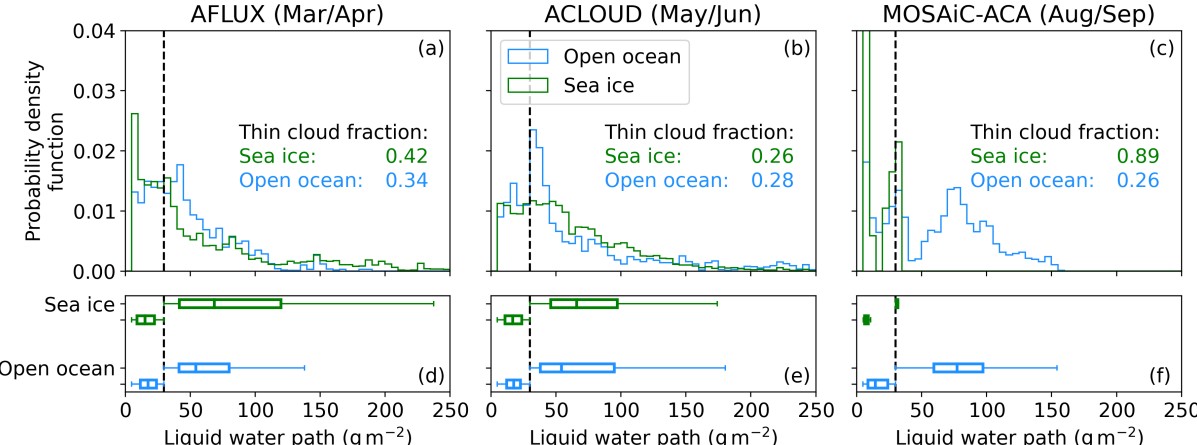

**Figure 3.** Probability density function of the equivalent LWP over sea ice and open ocean (colour-coded) for (a) AFLUX, (b) ACLOUD, and (c) MOSAiC-ACA. Only observations classified as cloudy (equivalent LWP $> 5\,\mathrm{g\,m^{-2}}$) were considered. The vertical dashed lines located at $30\,\mathrm{g\,m^{-2}}$ indicate the threshold used for the separation of thin and thick clouds. The numbers in (a–c) represent the fraction of observations with thin clouds with respect to the total amount of cloudy observations. (d–f) Box-whisker plots of the equivalent LWP separating thin (left boxplots) and thick clouds (right boxplots).

were thin. In contrast, the equivalent LWP of clouds over open ocean showed a broader distribution with a strong mode of thick clouds. The thin cloud fraction over open ocean was similar to the one observed during ACLOUD. Compared to the other campaigns, the median of the equivalent LWP of thin clouds over open ocean was slightly lower and amounted to $14\,\mathrm{g\,m^{-2}}$.

## 3  Simulation of the net irradiance in assumed cloud-free conditions

### 3.1  Radiative transfer simulations

The net irradiance was measured in cloudy conditions (Sect. 2.1). To calculate the CRE (Eq. 2), the net irradiance in cloud-free conditions needs to be simulated. For this purpose, the one-dimensional radiative transfer solver DISORT is applied (Stamnes et al., 1988), which is embedded in the library for radiative transfer *libRadtran* (Emde et al., 2016). The radiative transfer simulations were performed using the method proposed by Stapf et al. (2020). The atmospheric state was obtained from thermodynamic profiles measured during ascents or descents adjacent to the respective low level-section, which were topped by the temporally closest radiosounding. Aerosol properties were not considered in the simulations. The downward and upward irradiances in the TIR range were simulated using the thermodynamic profile and a surface emissivity of 0.99 for snow (Warren, 1982), which is similar for open ocean at least for the atmospheric window (8–13 $\mu$m) region (Konda et al., 1994). Beside the thermodynamic profile and the SZA, the simulation of the cloud-free irradiances in the solar spectral range requires to define the local surface albedo in cloud-free conditions, which is different from the surface albedo measured in cloudy conditions. From the simulated solar irradiances, the direct/diffuse fractions of the downward solar irradiance were obtained.



## 3.2 Impact of clouds on the surface albedo

### 200 3.2.1 Surface albedo over open ocean in cloud-free and cloudy conditions

While Stapf et al. (2020) analyzed the impact of clouds on the surface albedo of sea ice, the following analysis focuses on similar effects influencing the surface albedo of open ocean. The change of the broadband albedo due to the presence of clouds is a result of two effects: First, the changing illumination geometry (geometry effect), and, second, the changing spectral distribution of the downward solar irradiance (spectral weighting effect). In general, the broadband surface albedo $\alpha$ is given

by

$$\alpha = \frac{F_{\mathrm{sol}}^{\uparrow}}{F_{\mathrm{sol}}^{\downarrow}} = \frac{\int_{\mathrm{sol}} F_{\lambda}^{\uparrow}(\lambda)\,\mathrm{d}\lambda}{\int_{\mathrm{sol}} F_{\lambda}^{\downarrow}(\lambda)\mathrm{d}\lambda}. \tag{3}$$

The upward and downward solar irradiances, $F_{\mathrm{sol}}^{\uparrow}$ and $F_{\mathrm{sol}}^{\downarrow}$, are obtained by integrating the spectral upward and downward irradiances, $F_{\lambda}^{\uparrow}$ and $F_{\lambda}^{\downarrow}$, over the wavelengths $\lambda$ of the solar spectral range (indicated by $\int_{\mathrm{sol}}$). To introduce the spectral albedo $\alpha_{\lambda}$, $F_{\lambda}^{\uparrow}$ is replaced by $F_{\lambda}^{\downarrow}$ and $\alpha_{\lambda}$:

$$\alpha = \int_{\mathrm{sol}} \alpha_{\lambda}(\lambda) \cdot \frac{F_{\lambda}^{\downarrow}(\lambda)}{\int_{\mathrm{sol}} F_{\lambda}^{\downarrow}(\lambda)\,\mathrm{d}\lambda}\,\mathrm{d}\lambda. \tag{4}$$

Equation 4 shows that the broadband albedo represents a weighted average depending on the spectral albedo and the downward spectral irradiance, which serves as a weight function. While the spectral albedo changes due to the geometry effect, a change of the normalized weight function $w$, with

$$w = \frac{F_{\lambda}^{\downarrow}(\lambda)}{\int_{\mathrm{sol}} F_{\lambda}^{\downarrow}(\lambda)\,\mathrm{d}\lambda}, \tag{5}$$

describes the spectral weighting effect. Both effects are analyzed in the following.

The spectral surface albedo and the normalized weight function over open ocean were simulated with *libRadtran*, applying a parameterization of the directional reflection of open ocean surfaces with varying SZA and wind speed in $10\,\mathrm{m}$ altitude (Cox and Munk, 1954). To illustrate the effect of clouds on the surface albedo, simulations containing boundary layer clouds (400–600 m) with variable LWP and a fixed $r_{\mathrm{eff}}$ of $8\,\mu\mathrm{m}$ were performed.

The change of the spectral albedo due to different illumination geometries is shown in Fig. 4a. In cloud-free conditions, the spectral albedo is dominated by the reflection of the direct component of the incident spectral irradiance. Beside their different patterns, the spectral albedo is significantly larger for a SZA of 75°, which is representative for AFLUX and MOSAiC-ACA, compared to a SZA of 60°, representative for ACLOUD. The almost constant spectral albedo above $1000\,\mathrm{nm}$ is 0.20 for 75°, but only 0.07 for 60°. This difference is due to the enhanced specular reflection at the air–water interface for larger incident

angles (i. e., SZA), according to Fresnel's equations. In cloudy conditions (LWP of $80\,\mathrm{g\,m^{-2}}$), the spectral surface albedo is modified by the diffuse illumination geometry and reveals slightly lower values compared to the cloud-free case with a SZA of 60°. The best agreement between the spectral albedo in cloudy and cloud-free conditions was found for a SZA of 52° (not





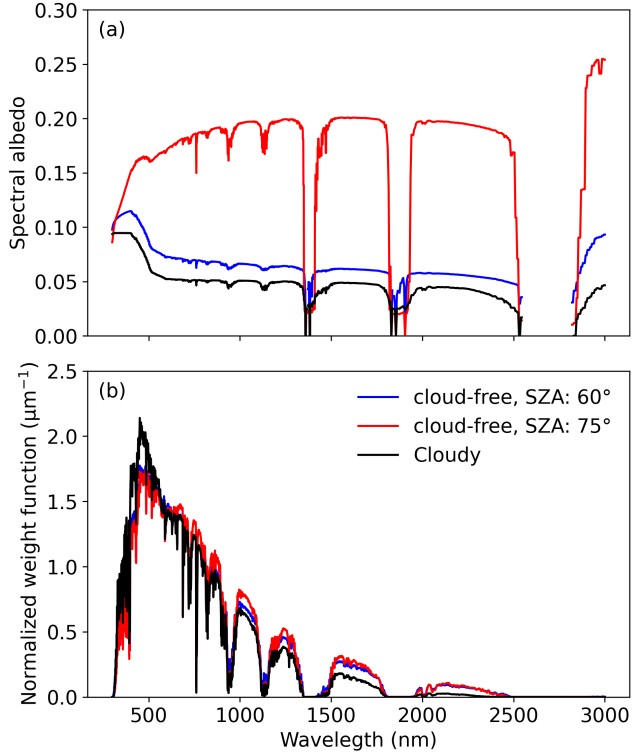

**Figure 4.** (a) Spectral surface albedo of open ocean for two different SZAs in cloud-free conditions, and for cloudy conditions (LWP of $80\,\mathrm{g\,m^{-2}}$, independent of SZA). (b) Spectral normalized weight functions $w$ for the same scenarios. The wind speed of $1\,\mathrm{m\,s^{-1}}$ represents a calm surface.

shown), which can be referred to as an effective incident zenith angle in cloudy conditions. Thus, the geometry effect causes a lower surface albedo in cloudy conditions compared to cloud-free conditions for SZAs typical for the Arctic, with a larger

difference for larger SZAs. This difference decreases as the wind speed is increased, especially for large SZAs (Jin et al., 2004).

The broadband albedo is also affected by the spectral distribution of the incident irradiance (Eq. 4), which is described by $w$ and can be modified by clouds. Spectra of $w$ corresponding to the cases discussed above are illustrated in Fig. 4b. At visible wavelengths (e. g., $500\,\mathrm{nm}$), $w$ is larger in cloudy conditions compared to cloud-free conditions, while the situation is reversed in the near-infrared (NIR) range (e. g., $1600\,\mathrm{nm}$) due to enhanced absorption by cloud particles. Consequently, the

spectral albedo in the visible wavelength range, which is slightly higher compared to the albedo in the NIR range, contributes more strongly to the broadband albedo in cloudy conditions than it does in the cloud-free case. This spectral weighting partly counteracts the spectral albedo geometry effect on the broadband albedo.





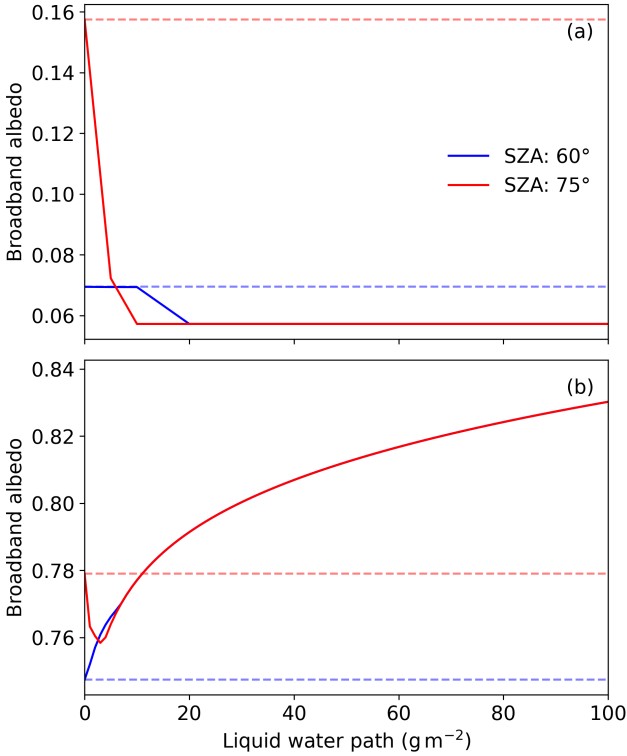

**Figure 5.** Broadband albedo of (a) open ocean, and (b) sea ice as a function of the cloud LWP for different SZAs. The open ocean albedo is based on the parameterization of Jin et al. (2011) with a 10 m wind speed of $1\,\mathrm{m\,s^{-1}}$. The sea ice albedo is based on the parameterization of Gardner and Sharp (2010) with a SSA of $80\,\mathrm{m^2\,kg^{-1}}$. The attenuated dashed lines represent the respective cloud-free surface albedos.

### 3.2.2 Parameterization of the open ocean and sea ice albedo in cloud-free conditions

To account for the cloud-induced change of the surface albedo in the calculation of the cloud-free net irradiance, parameteriza-
tions are used to retrieve the surface albedo in cloud-free conditions. The sea ice albedo is retrieved using the method described by Stapf et al. (2020), which is based on the parameterization of Gardner and Sharp (2010). This parameterization depends on the SZA, the equivalent cloud LWP, and the specific surface area (SSA) of snow, which is a measure for the snow grain size and was retrieved from the measured surface albedo. For open ocean, the parameterization of Jin et al. (2011) was used to obtain the surface albedo in cloud-free conditions. The required input include the SZA, the surface wind speed measured
in flight altitude and scaled down to 10 m using the logarithmic wind profile with a roughness length of $2 \cdot 10^{-4}\,\mathrm{m}$ (offshore conditions), and the simulated fraction of diffuse incident radiation in cloud-free conditions (Sect. 3.1). For a mixture of open ocean and sea ice, the parameterized albedos of both surface types are linearly combined using the sea ice concentration.

Figure 5 illustrates the parameterized broadband albedo as a function of the LWP. The broadband open ocean albedo (Fig. 5a) decreases with increasing LWP, which indicates that the geometry effect dominates over the spectral weighting effect. This is
due to the relatively low spectral differences of the spectral open ocean albedo (Fig. 4a). Similar to the spectral albedo, the



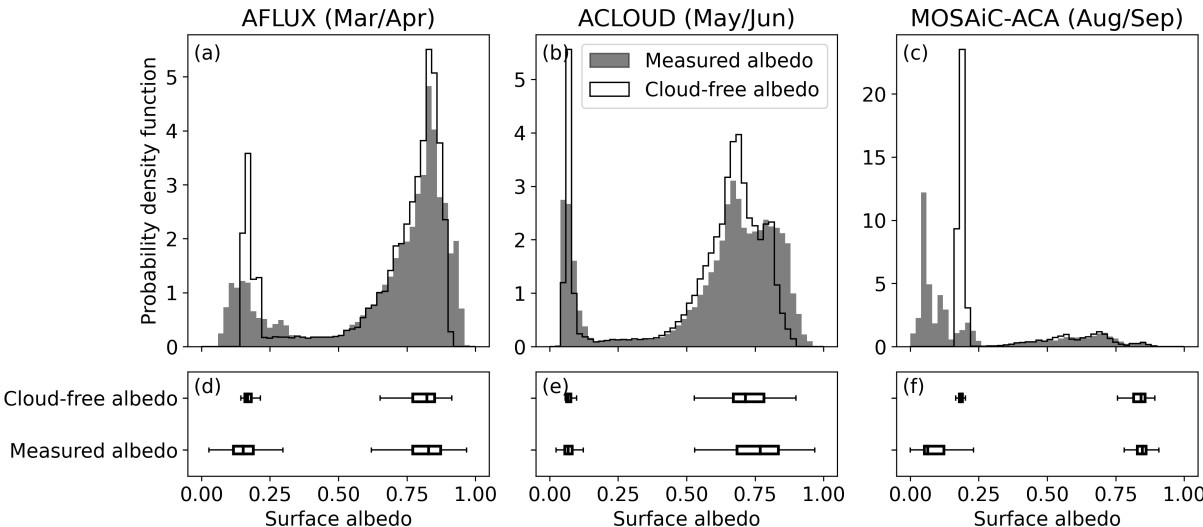

**Figure 6.** Probability density function of the surface albedos measured in mostly cloudy conditions (shadings) and retrieved for cloud-free conditions (lines) for (a) AFLUX, (b) ACLOUD, and (c) MOSAiC-ACA. (d–f) Box-whisker plots of the cloud-free and measured surface albedo separated for open ocean (left boxplots) and sea ice (right boxplots).

broadband open ocean albedo in diffuse conditions (LWPs larger than $20 \, \mathrm{g \, m^{-2}}$) is independent of the SZA and amounts to 0.06, while the albedo in cloud-free conditions increases for increasing SZA. Thus, the cloud-free albedo is only slightly larger than the diffuse albedo for a SZA of $60°$, but reaches 0.16 for a SZA of $75°$. For larger wind speeds, the difference between the cloudy and cloud-free albedos decreases (not shown).

In contrast to open ocean, the spectral albedo of snow-covered sea ice shows large discrepancies, with high values in the visible and low values in the NIR spectral range (e. g., Stapf et al., 2020, their Fig. 3). Thus, the spectral weighting effect becomes more dominant and leads to an increase of the broadband albedo with increasing LWP (Fig. 5b). While this is true for the entire LWP range for a SZA of $60°$, a slight albedo decrease is observed for the lowest LWPs and a SZA of $75°$. The cloud-free albedo is 0.75 and 0.78 for SZAs of $60°$ and $75°$, respectively. Similar to open ocean, the surface albedo of sea ice

does not differ with SZA in diffuse conditions. Consequently, the albedo differences between cloudy and cloud-free conditions are larger for lower SZAs within the typical LWP range.

      To retrieve the cloud-free surface albedo from the mostly cloudy albedo measurements performed during the low-level sections, the surface albedo parameterizations were applied to the measurements of the three campaigns. Figure 6 illustrates the effect of the cloud-induced albedo change by comparing the frequency distributions of the measured (cloudy) and retrieved

(cloud-free) surface albedos. The left and the right mode of all distributions correspond to the open ocean and sea ice surfaces, respectively. Compared to the measured albedo, the distribution of the cloud-free albedo is narrowed over open ocean during AFLUX (Figs. 6a, 6d). Although a notably higher open ocean albedo would be expected in cloud-free conditions for SZAs present during AFLUX (Fig. 5a), the cloudy and cloud-free median albedo values of 0.15 and 0.17, respectively, did not differ





significantly. This is probably due to the frequently occurring sea smoke between the aircraft and the ocean surface, which

artificially increased the measured albedo compared to the expected values (Fig. 5a). For the snow-covered sea ice, the median values of 0.83 and 0.82 indicate only small differences between the cloudy and cloud-free albedos, which is in accordance with Fig. 5b for the typical SZA range during AFLUX. Similarly, no significant difference between cloudy and cloud-free albedo of open ocean could be observed during ACLOUD (Figs. 6b, 6e). However, as suggested by Fig. 5b, a significant albedo shift over sea ice is obvious for ACLOUD, where the median of the surface albedo decreased from 0.77 in the observed cloudy

conditions to 0.71 for the retrieved cloud-free albedo. In general, the observations during ACLOUD are characterized by a lower sea ice albedo compared to AFLUX, which is due to the onset of the melting season. Despite the similar SZA range, the cloud-induced change of the open ocean albedo showed different effects for AFLUX and MOSAiC-ACA. The median of the measured cloudy albedo was only 0.06 during the latter (Figs. 6c, 6f). However, the median cloud-free albedo of 0.19 was similar to the value retrieved for AFLUX. The rarely sampled sea ice during MOSAiC-ACA is only expressed by the rightmost,

very small mode. Similar to AFLUX, the sea ice albedo did not change significantly between cloudy and cloud-free conditions. The mode located roughly between 0.25 and 0.75 represents observations over the MIZ.

## 4 Cloud radiative effect

### 4.1 Solar cloud radiative effect

The variability of the solar CRE at the surface during the three campaigns is assessed by analyzing their different mode

structures. Since the solar CRE strongly depends on the surface albedo, Fig. 7 shows the frequency distributions of the solar CRE as a function of the measured surface albedo. To quantify the impact of the cloud-induced albedo change on the solar CRE, two distributions are presented for each campaign. While the surface albedo change is neglected in Figs. 7a–7c, Figs. 7d–7f show the solar CRE corrected for this effect. For AFLUX and ACLOUD, the distributions of Fig. 7 reveal four distinct modes (indicated by the numbers 1–4). The modes 1 and 2 are located around or slightly above $0 \, \mathrm{W \, m^{-2}}$ and reflect mainly cloud-

free conditions, while the remaining modes (modes 3 and 4) indicate the cooling effect of clouds in the solar spectral range. Through their distinct surface albedos, the solar CREs over open ocean (mode 3) and sea ice (mode 4) surfaces are clearly distinct. During MOSAiC-ACA (Figs. 7d, 7f), thin or broken clouds (mode 5) and frequent observations over the MIZ in cloudy conditions (mode 6) altered the mode structure.

The comparison of the two distributions per campaign exposes the shift of several modes due to the cloud-induced albedo

change. For AFLUX (Figs. 7a, 7d), only minor changes can be observed, which is in accordance with the similar distributions of measured and retrieved (cloud-free) surface albedo (Fig. 6a). The albedo change increased (reduced) the median solar CRE over open ocean (sea ice) by $4 \, \mathrm{W \, m^{-2}}$. The artificial absence of a larger CRE increase over open ocean is due to the sea smoke and the resulting too high measured surface albedo discussed in Sect. 3.2.2. The actual change of the solar CRE remains unclear. As discussed by Stapf et al. (2020), the increased albedo of sea ice in cloudy conditions caused a significant shift of

mode 4 during ACLOUD (Figs. 7b, 7e) and almost doubled the median of the uncorrected solar cooling effect of $-33 \, \mathrm{W \, m^{-2}}$. In contrast, the solar CRE over open ocean was hardly affected. During MOSAiC-ACA, the reduction of the solar cooling effect

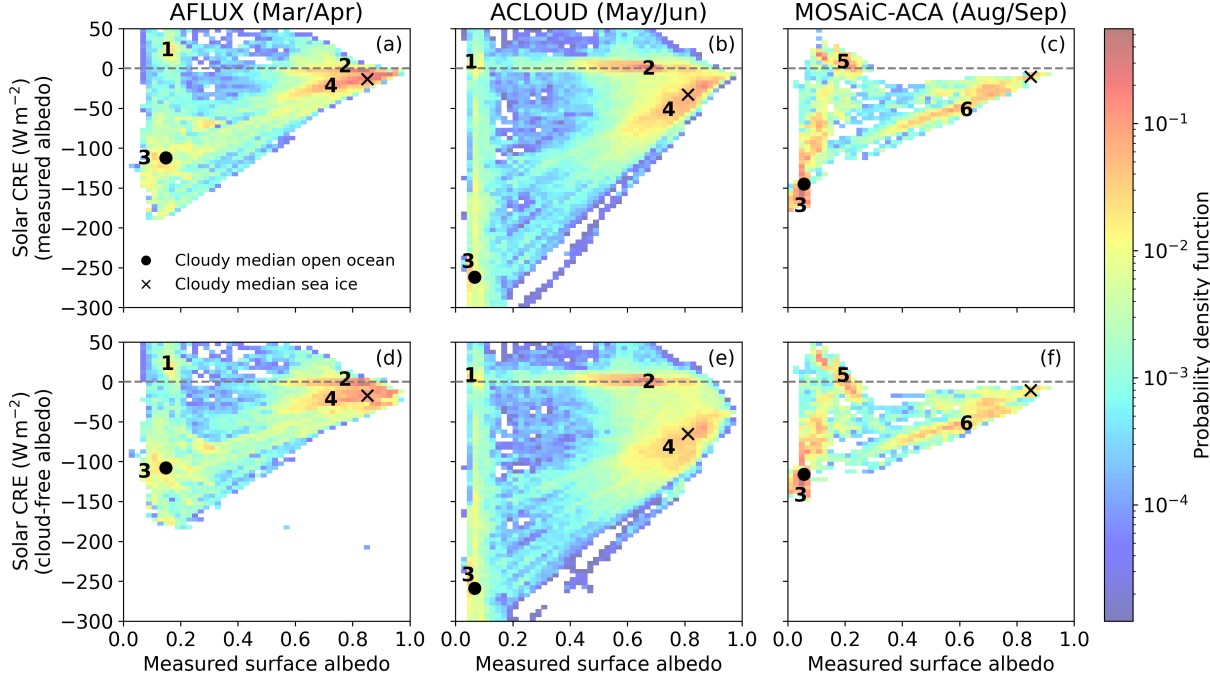

**Figure 7.** Two-dimensional probability density function of the solar CRE and the measured (mostly cloudy) surface albedo (as an indicator for the surface type) for AFLUX (a,d), ACLOUD (b,e), and MOSAiC-ACA (c,f). The upper row (a–c) shows the distributions of the solar CRE neglecting the cloud-induced change of the surface albedo, while this effect is included in the lower row (d–f). The horizontal dashed lines mark a CRE of $0\,\mathrm{W\,m^{-2}}$. The symbols represent the cloudy (equivalent LWP $> 5\,\mathrm{g\,m^{-2}}$) median CREs and measured surface albedos over the different surface types.

due to the increased open ocean albedo in cloud-free conditions (Fig. 6c) is expressed by the shift of mode 3 (Figs. 7c, 7f). The median of the uncorrected solar CRE ($-145\,\mathrm{W\,m^{-2}}$) was increased by $29\,\mathrm{W\,m^{-2}}$. The sea ice–dominated MIZ (mode 6) is not affected by the albedo change, because the slight increase of the sea ice albedo and the more significant decrease of the open ocean albedo towards cloud-free conditions cancel.

Using the solar CRE accounting for the cloud-induced albedo change (Figs. 7d–7f), the features of the individual distributions and the differences among them are discussed in the following. Interestingly, the cloud-free open ocean mode (mode 1) of the AFLUX distribution (Fig. 7d) shows a positive CRE, with a median of $20\,\mathrm{W\,m^{-2}}$ for an equivalent LWP of less than $5\,\mathrm{g\,m^{-2}}$. Probably, broken clouds, which in most cases do not shade the Sun, caused an enhanced solar downward irradiance compared to the cloud-free radiative transfer simulations, by scattering additional solar radiation towards the surface (Schade et al., 2007). This might have led to a retrieved equivalent LWP, which is low and, thus, almost indistinguishable from cloud-free conditions. Due to the high surface albedo of sea ice (larger than 0.6), the magnitude of the solar CRE was small over this surface type (mode 4, median of $-17\,\mathrm{W\,m^{-2}}$) and could be hardly distinguished from the absent CRE in cloud-free conditions





(mode 2), especially for very bright scenes. In contrast, the low open ocean albedo enabled a much larger solar cooling effect

of the clouds (mode 3) with a median CRE of $-108\,\mathrm{W\,m^{-2}}$.

The CRE distribution of ACLOUD (Fig. 7e) is shaped more clearly compared to AFLUX due to the better statistics of the data. The solar CRE over open ocean and sea ice (modes 3 and 4) reveals median values of $-259\,\mathrm{W\,m^{-2}}$ and $-65\,\mathrm{W\,m^{-2}}$, respectively, which indicates a significantly larger magnitude of the solar cooling effect compared to AFLUX. Although the lower surface albedo contributed to the larger cooling effect during ACLOUD, the major contribution to the solar CRE differ-

ences between the two campaigns resulted from the different solar illumination as a consequence of the clearly distinct SZA ranges (Table 1). Note that the SZA exhibits not only an annual, but also a daily cycle. However, since most of the flights were conducted around solar noon, the SZA variability within one campaign was small.

The observations from MOSAiC-ACA reveal a large variability and a relatively blurry mode structure (Fig. 7f), which is due to the much less significant statistics compared to the other campaigns (see Table 1). Modes 1 and 2 are missing in

the distribution, since cloud-free conditions were not sampled during MOSAiC-ACA. Instead of mode 1, mode 5 represents broken or very thin clouds over open ocean. Similar to AFLUX, this mode peaks at positive CRE values due to the broken cloud effect. However, in contrast to AFLUX, also negative CRE values were observed, probably resulting from rather overcast thin cloud conditions. Due to the similar Sun elevation (Table 1), the median of the CRE over open ocean ($-116\,\mathrm{W\,m^{-2}}$, mode 3) was similar compared to AFLUX. In contrast to the other campaigns, the rare observations over homogeneous sea ice during

MOSAiC-ACA are not reflected in an own mode. These observations are rather included in the cloudy MIZ mode (mode 6), which expands to albedo values of down to 0.4.

The analysis of the solar CRE indicated a significant variability of the solar CRE between the campaigns and the underlying surface types. In accordance with previous studies (e. g., Intrieri et al., 2002; Miller et al., 2015), the solar CRE between the campaigns mostly varied due to the seasonally changing SZA, with an increasing cooling effect for a decreasing SZA. The

differences between open ocean and sea ice are a result of their distinct surface albedos. The impact of clouds on the surface albedo affects the solar CRE differently, depending on surface type and SZA. While the albedo in cloudy conditions and the solar cooling effect of clouds over sea ice are mostly increased for relatively low Arctic SZAs, larger SZAs rather reduce the cooling effect over open ocean. Given the seasonality of the sea ice extent in the Fram Strait region (Fig. 1), this effect rather enhances the cooling effect during early summer and reduces it during late summer. The resulting convergence of the

affected cloudy modes during ACLOUD and MOSAiC-ACA (Fig. 7) suggests a reduced amplitude of the seasonal CRE cycle considering the cloud-induced albedo modifications.

## 4.2 Thermal-infrared cloud radiative effect

The TIR CRE is determined by a complex interplay of the environmental thermodynamics, and the cloud macrophysical and microphysical properties. The frequency distributions of the TIR CRE for all three campaigns are depicted in Fig. 8, separated

for sea ice and open ocean. Independent of the underlying surface type, the distributions of the TIR CRE during AFLUX (Figs. 8a, 8d) and ACLOUD (Figs. 8b, 8e) reveal two distinct modes. Similar to the solar CRE, the mode located around $0\,\mathrm{W\,m^{-2}}$ represents cloud-free conditions, while the second mode clearly indicates the warming effect of the clouds in the TIR





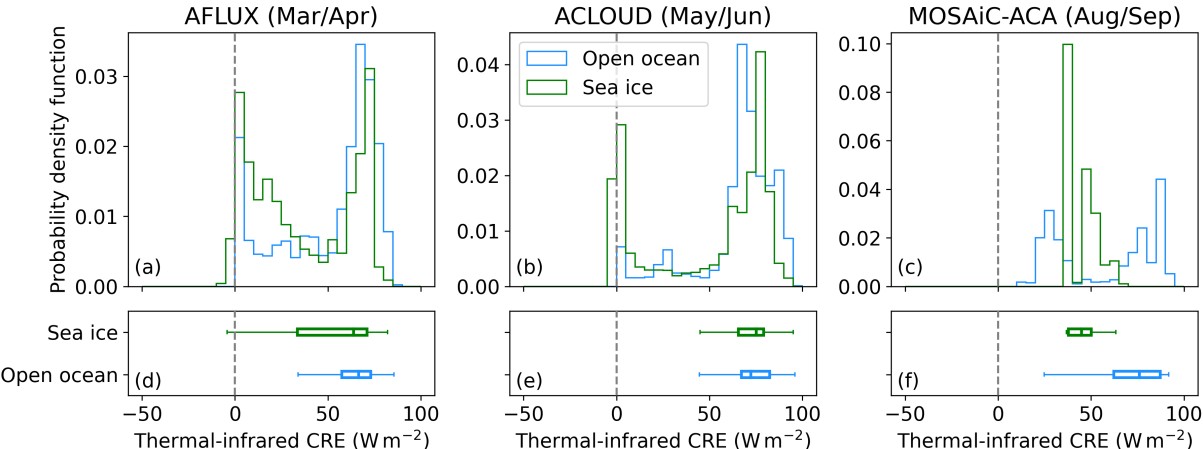

**Figure 8.** Probability density function of the TIR CRE for (a) AFLUX, (b) ACLOUD, and (c) MOSAiC-ACA, separated for sea ice and open ocean (colour-coded). (d–f) Box-whisker plots of the TIR CRE only considering cloudy observations (equivalent LWP $> 5\,\mathrm{g\,m^{-2}}$). The vertical dashed lines mark a CRE of $0\,\mathrm{W\,m^{-2}}$.

range. In contrast to the solar CRE, the order of magnitude of the TIR CRE range does not differ between the surface types because the surface temperature does not affect the CRE. Differences result only from the influence of the surface on the cloud

and thermodynamic properties.

The median TIR CRE observed in cloudy conditions during AFLUX was $67\,\mathrm{W\,m^{-2}}$ over open ocean and $64\,\mathrm{W\,m^{-2}}$ over sea ice. The slightly lower value for the latter was caused by the enhanced frequency of observations with relatively low TIR CRE ($< 30\,\mathrm{W\,m^{-2}}$), which might be linked to the larger fraction of thin clouds (Fig. 3a) or to the slightly lower cloud base temperature (Fig. 2) over sea ice. For AFLUX, a cloud base temperature change of $1\,\mathrm{K}$ results in a change of the TIR CRE in

the order of $4\,\mathrm{W\,m^{-2}}$, which is approximately the difference between the median TIR CREs over sea ice and open ocean.

During ACLOUD (Figs. 8b, 8e), cloud-free conditions were significantly less frequent over open ocean than over sea ice. Instead, the distribution of open ocean reveals an additional small mode around $25\,\mathrm{W\,m^{-2}}$. The TIR CRE in cloudy conditions did not differ significantly between the surface types with median values ranging between $72\,\mathrm{W\,m^{-2}}$ over open ocean and $75\,\mathrm{W\,m^{-2}}$ over sea ice. The slightly larger TIR CRE over sea ice cannot be explained by the cloud base temperature (Fig. 2),

which showed lower values over sea ice. Likely, the more frequent occurrence of thick clouds (Fig. 3b) caused this tendency. Also compared to AFLUX, the thicker clouds during ACLOUD are likely the reason for the larger surface TIR CRE, since the increased absolute humidity counteracts the effect of the higher cloud base temperature during ACLOUD (Cox et al., 2015). Except the additional small mode for open ocean, the cloud-free and cloudy modes are more clearly separated during ACLOUD compared to AFLUX.

Similar to the solar CRE, the low amount of data obtained during MOSAiC-ACA results in a less significant mode structure (Figs. 8e, 8f). Only in the distribution of open ocean, two distinct modes are visible. Due to the lack of cloud-free observations, the mode with the smallest CRE is located around $25\,\mathrm{W\,m^{-2}}$ and represents the broken cloud conditions. However,





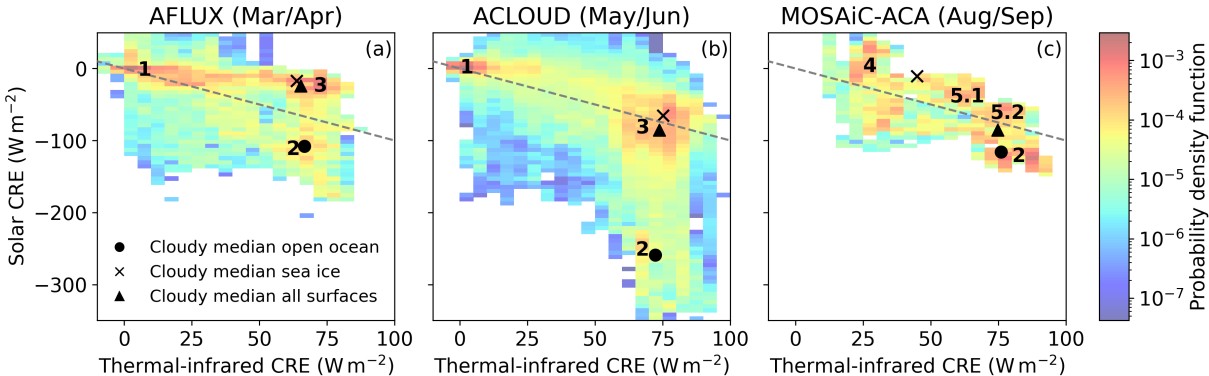

**Figure 9.** Two-dimensional probability density function of the solar and the TIR CRE for (a) AFLUX , (b) ACLOUD, and (c) MOSAiC-ACA. The diagonal dashed lines represent the $0\,\mathrm{W\,m^{-2}}$ isoline of the total (i. e., the sum of the solar and TIR) CRE. The symbols represent the cloudy (equivalent LWP $> 5\,\mathrm{g\,m^{-2}}$) median CREs over the different surface types.

these observations mostly coincide with an equivalent LWP of less than $5\,\mathrm{g\,m^{-2}}$ and do not significantly contribute to the cloudy median CRE of $75\,\mathrm{W\,m^{-2}}$ over open ocean. This median value is similar to the median TIR CRE obtained during

ACLOUD, which is probably due to the similar frequency of thick clouds (Fig. 3). Over sea ice, a significantly lower TIR CRE was observed during MOSAiC-ACA ($45\,\mathrm{W\,m^{-2}}$), which results from the limited sampling statistics. Nevertheless, this is in agreement with the lower observed cloud base temperature (Fig. 2) and the significantly thinner clouds (Fig. 3c).

### 4.3 Total cloud radiative effect

The previous analysis showed that the variability of the CRE between the campaigns and the surface types is significantly lower

in the TIR than in the solar spectral range. Thus, the variability of the solar CRE is the major driver of the variability of the total (i. e., the sum of solar and TIR) CRE. Depending on whether the solar cooling or the TIR warming effect dominates, the total CRE determines whether a cloud has a cooling or a warming effect on the surface. Figure 9 illustrates two-dimensional frequency distributions combining the solar and the TIR effects to assess the total CRE.

All modes visible in Fig. 9 can be attributed to the modes discussed in Figs. 7 and 8. In the distributions of AFLUX (Fig. 9a)

and ACLOUD (Fig. 9b), mode 1 is clustered around $0\,\mathrm{W\,m^{-2}}$ for both solar and TIR CRE and combines the cloud-free observations over open ocean and sea ice. The clearly distinct solar cooling effect over the different surface types (Fig. 7) separates the cloudy modes over open ocean (mode 2, larger solar cooling effect) and sea ice (mode 3), while the TIR warming effect is similar.

Over open ocean (mode 2), clouds of sufficient LWP showed a total cooling effect (values below the dashed line in Fig. 9)

during all campaigns. However, the magnitudes of the CRE differed significantly as quantified by the median values of $-48\,\mathrm{W\,m^{-2}}$, $-185\,\mathrm{W\,m^{-2}}$, and $-36\,\mathrm{W\,m^{-2}}$ during AFLUX, ACLOUD, and MOSAiC-ACA, respectively. Due to the lower SZA during ACLOUD, the solar cooling effect dominated the total CRE. Over sea ice, the TIR warming effect was dominant





over the solar cooling effect during AFLUX, resulting in a median total warming of $42\,\mathrm{W\,m^{-2}}$. During ACLOUD, however, the solar cooling roughly compensated the TIR warming, leading to a small median total CRE of $7\,\mathrm{W\,m^{-2}}$.

During AFLUX, a significant amount of observations ranges in a transition between modes 1 and 3. These data correspond to the observations over sea ice with low TIR CRE that were already discussed in Fig. 8a. The clouds during these situations were rather thin with a median equivalent LWP of $19\,\mathrm{g\,m^{-2}}$ compared to $54\,\mathrm{g\,m^{-2}}$ for the observations forming the cloudy sea ice mode (mode 3). The almost absent solar cooling in combination with a TIR warming leads to the total warming effect of these thin clouds often discussed in literature (e. g., Miller et al., 2015), which under certain circumstances can induce ice

melting (Bennartz et al., 2013).

Again, the sparse statistics during MOSAiC-ACA need to be interpreted with caution and represent only a subsample of possible cloud conditions (Fig. 9c). Instead of the cloud-free mode (mode 1) observed for the other campaigns, mode 4 represents the broken clouds that produced only a slightly positive solar, but a significantly positive TIR CRE with median values of $7\,\mathrm{W\,m^{-2}}$ and $26\,\mathrm{W\,m^{-2}}$, respectively. The resulting positive total CRE underlines that the warming effect of broken clouds

can be observed also over open ocean. Modes 5.1 and 5.2 reveal a cloud warming effect over the MIZ, while a separate sea ice mode is missing.

For the regional average CRE, the sea ice concentration within the area of observations is most relevant. While the sea ice situation in the Fram Strait north-west of Svalbard was similar during AFLUX and ACLOUD (Figs. 1a and 1b), the sea ice edge was located significantly further north during MOSAiC-ACA (Fig. 1c) and the MIZ was broader. This also affected the

fraction of observations obtained for each of the surface types and determined the strength of the different modes in Fig. 9. As a proxy for the entire Fram Strait, the median of the total CRE observed in cloudy conditions during one single campaign was calculated regardless of surface type (triangles in Fig. 9). For MOSAiC-ACA, this median CRE of $-27\,\mathrm{W\,m^{-2}}$ was close to the median for open ocean due to the dominance of this surface type. During AFLUX, a positive median CRE of $25\,\mathrm{W\,m^{-2}}$, and during ACLOUD, a negative median CRE of $-16\,\mathrm{W\,m^{-2}}$ was observed, both ranging close to median values of sea ice.

Given the different SZA ranges during both campaigns, this indicates that clouds over the Fram Strait showed a warming effect on the surface during AFLUX (spring) and an almost neutral to slightly negative CRE during ACLOUD (early summer). MOSAiC-ACA revealed a similar median of the total CRE compared to AFLUX for the individual surface types. However, the dominant open ocean surfaces during MOSAiC-ACA caused the clouds to have an average cooling effect during late summer.

## 5   Conclusions

To compare the warming or cooling effects of clouds over sea ice and open ocean during different times of the year, the surface CRE was evaluated from observations performed during three campaigns in the Fram Strait north-west of Svalbard. The campaigns AFLUX, ACLOUD, and MOSAiC-ACA were characterized by significantly different sea ice coverages and thermodynamic states during spring, early summer, and late summer, respectively. The CRE was calculated from a combination of airborne broadband radiation measurements and radiative transfer simulations. While the net irradiance in cloudy conditions

was measured during low-level flight sections, the corresponding cloud-free net irradiance was simulated and accounted for





changes of the surface albedo between cloudy and cloud-free conditions (Stapf et al., 2020). This was done by retrieving the cloud-free surface albedo from parameterizations for sea ice and open ocean surfaces. The radiative impact of clouds on the surface albedo differed between the surface types and was not uniform among the different campaigns.

The solar and total CRE were affected distinctly by these cloud-induced differences of the surface albedo, mainly depending on the SZA. The consideration of this effect almost doubled the solar cooling effect over sea ice during ACLOUD, as already discussed by Stapf et al. (2020). In contrast, the larger SZAs present during the other campaigns suppressed similar changes of the CRE. However, over open ocean, the impact of the cloud-induced albedo change increases with increasing SZA. Thus, the solar cooling effect over open ocean was reduced by about 20 % during MOSAiC-ACA, while ACLOUD was not affected. During AFLUX, the albedo change was masked by the presence of sea smoke.

The solar CRE strongly varied between sea ice and open ocean surfaces as well as among the campaigns. While the low surface albedo caused a large solar cooling effect over open ocean, the cooling effect over sea ice was rather weak and partly not distinguishable from cloud-free conditions. This weak cooling was often supported by the presence of thin clouds. Due to the lower SZA, the solar CRE showed a significantly larger cooling during ACLOUD compared to the other campaigns.

    The variability of the TIR CRE results from a complex interplay between changing thermodynamic and cloud properties. In
contrast to the solar CRE, the TIR CRE varied only weakly between the surface types and the campaigns and mostly showed median values between $64\,\mathrm{W\,m^{-2}}$ and $76\,\mathrm{W\,m^{-2}}$. Compared to the other campaigns, a lower TIR CRE was found during AFLUX, which was caused by the enhanced frequency of optically thin clouds.

    The variability of the total CRE is driven by the solar CRE. A total cooling effect, dominated by the solar cooling effect, was found over open ocean during all campaigns. This cooling effect was largest during ACLOUD ($-185\,\mathrm{W\,m^{-2}}$) compared
to the other two campaigns (around $-40\,\mathrm{W\,m^{-2}}$). Over sea ice and the MIZ, the TIR warming effect clearly dominated during AFLUX ($42\,\mathrm{W\,m^{-2}}$) and MOSAiC-ACA ($22\,\mathrm{W\,m^{-2}}$), while the solar cooling and the TIR warming effect roughly compensated during ACLOUD. Broken and optically thin clouds showed a total warming effect, independent of the underlying surface. This is due to their almost neutral to slightly positive solar CRE and their significantly positive TIR CRE. In addition to the SZA, the total CRE not separated for the surface types differs between the campaigns due to the seasonally different sea ice
distributions. For each campaign, the sea ice distribution in the Fram Strait region is imprinted in the fraction of observations over the respective surface types. The high fraction of observations over sea ice during AFLUX and ACLOUD implies a warming and almost neutral effect of clouds in the Fram Strait during spring and early summer, respectively. In contrast, the frequent observations over open ocean during MOSAiC-ACA cause a cooling effect of clouds in the Fram Strait during late summer.

The short low-level flight segments during the campaigns are not necessarily representative for an entire season from a climatological point of view. However, various CRE differences between the campaigns could be attributed to their seasonality. Although observations of annual cycles of the CRE over sea ice are available (e. g., SHEBA, MOSAiC), the lack of long-term observations over open ocean complicates a robust characterization of the CRE. This especially holds since ice-free conditions will likely become more dominant in the future Arctic. However, the predominant cooling effect of clouds over open ocean will
lead to a negative contribution to further warming in the Arctic. This study and the published datasets of the CRE in the Fram



Strait (Stapf et al., 2021c; **?**) provided a basis for further investigations of cloud-related processes and feedback mechanisms in numerical models.

*Data availability.* All data analyzed in this manuscript are published on the PANGAEA database. The broadband irradiance and KT19 data can be found at Stapf et al. (2019, https://doi.pangaea.de/10.1594/PANGAEA.900442, ACLOUD), Stapf et al. (2021b, https://doi. pangaea.de/10.1594/PANGAEA.932020, AFLUX), and Becker et al. (2021b, https://doi.pangaea.de/10.1594/PANGAEA.936232, MOSAiC-ACA). The meteorological measurements (temperature, RH, wind speed) during the flights were published by Hartmann et al. (2019, https://doi.pangaea.de/10.1594/PANGAEA.902849, ACLOUD), Lüpkes et al. (2022, https://doi.pangaea.de/10.1594/PANGAEA.945844, AFLUX), and Hartmann et al. (2022, https://doi.pangaea.de/10.1594/PANGAEA.947787, MOSAiC-ACA). Dropsonde measurements were provided by Ehrlich et al. (2019a, https://doi.pangaea.de/10.1594/PANGAEA.900204, ACLOUD), Becker et al. (2020, https://doi.pangaea. de/10.1594/PANGAEA.921996, AFLUX), and Becker et al. (2021a, https://doi.pangaea.de/10.1594/PANGAEA.933581, MOSAiC-ACA), while the radiosounding are available at Maturilli (2020, https://doi.pangaea.de/10.1594/PANGAEA.914973). The microphysical cloud properties obtained from the in situ cloud probes can be found at Dupuy et al. (2019, https://doi.pangaea.de/10.1594/PANGAEA.899074, ACLOUD), Moser and Voigt (2022, https://doi.pangaea.de/10.1594/PANGAEA.940564, AFLUX), and Moser et al. (2022, https://doi. pangaea.de/10.1594/PANGAEA.940557, MOSAiC-ACA). Datasets containing the retrieved sea ice fraction, equivalent LWP, cloud-free albedo, and CRE were published by Stapf et al. (2021c, https://doi.pangaea.de/10.1594/PANGAEA.932010, ACLOUD and AFLUX) and **?**, ..., MOSAiC-ACA.

*Author contributions.* All authors contributed to the discussion of the results and the editing of the article. SB drafted the article, analyzed the data, and performed the ocean albedo analysis. MW and AE designed the experimental basis of this study.

*Competing interests.* The authors declare that they have no conflict of interest.

*Acknowledgements.* We gratefully acknowledge the funding by the Deutsche Forschungsgemeinschaft (DFG, German Research Foundation) – Projektnummer 268020496 – TRR 172, within the Transregional Collaborative Research Center "ArctiC Amplification: Climate Relevant Atmospheric and SurfaCe Processes, and Feedback Mechanisms (AC)[3]. This work was carried out and data used in this article were produced as part of the international Multidisciplinary drifting Observatory for the Study of Arctic Climate (MOSAiC) with the tag MOSAiC20192020 during the Airborne observations in the Central Arctic (MOSAiC-ACA, P5-221_MOSAiC_ACA_2020). We thank AWI logistics department, the crews of the research aircraft Polar 5 and 6 and all persons involved in the expedition of the Research Vessel Polarstern during MOSAiC (AWI_PS122_00) as listed in Nixdorf et al. (2021). This work was funded by the Open Access Publishing Fund of Leipzig University supported by the DFG within the program Open Access Publication Funding.



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
