# Peer review of "Airborne observations of the surface cloud radiative effect during different seasons over sea ice and open ocean in the Fram Strait"

_Atmospheric Chemistry and Physics, 2022_

## Author Comment (AC1)

**Answers to comments of Review #1**

We gratefully thank the reviewers for the positive feedback on our submitted manuscript. We appreciate the time they took to extensively read and comment on the given manuscript. The constructive comments are very helpful for the improvement of the manuscript. Our replies to the referees' comments are structured as follows:

*Referee's comments in italic – line numbers according to initially submitted manuscript*
Authors' responses in roman – line numbers according to adjusted manuscript. **Citations from the initial and the adjusted manuscript are given in bold.**

*The discussion of the important terms is inconsistent and sometimes confusing in the text. I recommend distinguishing explicitly between LW (TIR) and SW terms throughout. This would start by expanding eqs (1) and (2) to define separate terms for LW and SW using subscripts and then referring to those terms as well as their summations, $CRE_{SW}$, $CRE_{LW}$, $CRE_{total}$ explicitly throughout the text.*

We agree that, sometimes, the terms were used in a slightly confusing way. E. g., we sometimes only referred to the solar or TIR CRE as CRE without specifying it further. To overcome this issue, we introduced the terms $CRE_{sol}$, $CRE_{TIR}$ and $CRE_{tot}$ to refer to the solar, TIR and the total CRE. Therefore, we also split our equations into two as suggested. We changed the text around equations 1 and 2 as follows:

"**The surface REB is investigated separately for the solar and TIR spectral ranges and quantified by the solar and TIR net irradiances, $F_{net,sol}$ and $F_{net,TIR}$, respectively. The net irradiances are defined as the difference of the respective downward ($F_{sol}^{\downarrow}$ and $F_{TIR}^{\downarrow}$) and upward ($F_{sol}^{\uparrow}$ and $F_{TIR}^{\uparrow}$) irradiances:**

(new equation 1): $F_{net,sol} = F_{sol}^{\downarrow} - F_{sol}^{\uparrow}$**

(new equation 2): $F_{net,TIR} = F_{TIR}^{\downarrow} - F_{TIR}^{\uparrow}$**

**The cloud impact on the REB is quantified by the solar and TIR cloud radiative effect (CRE), $CRE_{sol}$ and $CRE_{TIR}$, respectively, which is also referred to as cloud radiative forcing (Ramanathan et al., 1989). It is derived from the net irradiances in cloudy ($F_{net,sol,cld}$ and $F_{net,TIR,cld}$) and cloud-free ($F_{net,sol,cf}$ and $F_{net,TIR,cf}$) atmospheric conditions:**

(new equation 3): $CRE_{sol} = F_{net,sol,cld} - F_{net,sol,cf}$**

(new equation 4): $CRE_{TIR} = F_{net,TIR,cld} - F_{net,TIR,cf}$**

**The sum of $CRE_{sol}$ and $CRE_{TIR}$ gives the total CRE $CRE_{tot}$. It depends on both …**"

Additionally, we now introduce the terms $CRE_{sol}$, $CRE_{TIR}$ and $CRE_{tot}$ already in the abstract and replaced "solar CRE", "TIR CRE" and "total CRE" as well as the unclear occurrences of "CRE", but also, e. g., "upward irradiance" or "downward irradiance" by the respective symbols throughout the text. However, when comparing two CRE values, we mostly stuck to the terms "solar cooling effect" or "TIR warming effect" since the wording "decreasing $CRE_{sol}$" instead of "stronger solar cooling effect" sounds counterintuitive.

[Figure]

Figure 1: Difference of $CRE_{TIR}$ (upper panels) and $CRE_{sol}$ (lower panels) between a flight altitude of 100 m and the surface for the different campaigns, depending on cloud base height, albedo, and LWP. (Figure not included in the manuscript)

*The CRE presented are advertised as referenceable to the surface but I don't see any description of the atmospheric corrections necessary to transfer the SW and LW observations from the altitude of the aircraft to the surface. If you think that the aircraft was low enough that no correction is needed, some evidence that the flux divergence between the aircraft and the surface is negligible is warranted.*

You are right that the CRE we observe at flight altitude differs from the real surface CRE due to the remaining atmospheric influence between the aircraft and the surface. To quantify this effect, we performed additional radiative transfer simulations for both $CRE_{sol}$ and $CRE_{TIR}$ at the surface and in 100 m altitude (the majority of the low-level measurements were performed below this altitude). In general, the simulation setup was identical to the description in the manuscript. Notwithstanding, for each campaign, the mean thermodynamic profile obtained during aircraft ascents and descents was used as input for both spectral ranges. For $CRE_{sol}$, the surface albedo was varied between 0 and 1 and a 200 m thick cloud based at 400 m was included with different LWP. The SZA was set to the mean campaign values. In the TIR range, 250 m thick clouds with different base heights and LWPs were added. The differences between the CRE at the surface and in 100 m are shown in Fig. 1. It turns out that $CRE_{sol}$ in this altitude is slightly underestimated compared to the surface level, while $CRE_{TIR}$ is slightly overestimated. For the situations present in our observations, the underestimation of $CRE_{sol}$ never exceeded 1.3 W m-2. The overestimation of $CRE_{TIR}$ does not exceed 1.25 W m-2. Consequently, we conclude that the CRE derived in low flight altitudes is a good estimate of the surface CRE. Cloud, surface and atmospheric properties will impact the CRE similarly in both altitudes.

Sometimes sea smoke was present in the lowest 100 m below the aircraft. In this case, a significantly different CRE could be expected at the surface. However, the focus in this study is on the radiative impact of clouds above the flight altitude. A correction of the sea smoke effects and transferring the CRE

derived in flight altitude to the surface CRE would mislead the interpretation of the CRE of the elevated clouds. Therefore, we stick to analyzing the CRE in flight altitude.

We added the following sentences to the text: "**Although the measurements were not performed directly at the surface, the impact of the atmosphere below the aircraft is small if no cloud or fog layers are present there. For a flight altitude of 100 m, radiative transfer simulations for different cloud and albedo properties revealed an underestimation of less than 1.3 W m$^{-2}$ for $CRE_{\mathrm{sol}}$ and an overestimation of less than 1.25 W m$^{-2}$ for $CRE_{\mathrm{TIR}}$ compared to the surface. Larger differences are expected for the occasionally occurring sea smoke below the aircraft. However, the analysis focuses on the radiative effect of clouds above the flight altitude, i. e., neglecting the sea smoke. Only in case of indirect effects (e. g., change of the measured albedo by the sea smoke), its influence is discussed in the remainder of this study.**"

*Line 310: I agree that locally and briefly downwelling shortwave at the surface can exceed TOA irradiance and could cause real positive values of solar CRE. However, I'm skeptical (in particular given the altitude of the aircraft) that this is such a significant effect on the surface values so as to make up as large a fraction of the samples as you show in Figure 7. Indeed, mode 1 ("cloud free"!) appears to be associated with positive solar CRE almost all the time. Something is amiss. Maybe you could validate your simulated clear-sky SW with observed SW under clear skies to be sure that (a) there is not a bias and (b) to potentially explain the preponderance of positive values as uncertainty in the clear term. If there are not enough statistics from the campaigns, perhaps a longer record of validation from Ny-Alesund can be performed.*

Just to avoid confusion and to clarify: We don't claim that, for the situations with positive $CRE_{\mathrm{sol}}$, the measured $F_{\mathrm{sol}}^{\downarrow}$ exceeds $F_{\mathrm{sol}}^{\downarrow}$ at TOA. Rather, we compare to simulated conditions (including an atmosphere), where simply no cloud is present. Anyway, as we agreed, broken clouds can briefly exceed the downward irradiance, which, however, does not depend on altitude.

Positive $CRE_{\mathrm{sol}}$ by broken and thin clouds is reported to be a common feature (e. g., Mol et al., 2023, https://doi.org/10.1029/2022JD037894). For our observations, it has to be noted that only two flights (31 March and 8 April, Fig. 2) contributed to the positive values of the $CRE_{\mathrm{sol}}$ distribution over open ocean (albedo values below 0.2, mode 1). To illustrate the occurrence of positive $CRE_{\mathrm{sol}}$ in more detail, time series of the measured and the cloud-free simulated $F_{\mathrm{sol}}^{\downarrow}$, the measured and retrieved cloud-free albedo and the obtained $CRE_{\mathrm{sol}}$ are shown in Fig. 3 for 8 April. For the first 30 minutes, the measured and simulated $F_{\mathrm{sol}}^{\downarrow}$ agreed well, which resulted in a $CRE_{\mathrm{sol}}$ of 0 W m$^{-2}$ and indicates that there is no major discrepancy between simulation and observation in cloud-free conditions. Roughly between 11:15 and 11:35, the measured $F_{\mathrm{sol}}^{\downarrow}$ was highly variable and partly exceeded the cloud-free simulated $F_{\mathrm{sol}}^{\downarrow}$, which led to the frequently observed positive $CRE_{\mathrm{sol}}$ values. The median solar CRE was 16 W m$^{-2}$. During

[Figure]

*Figure 2: Scatter plot of the $CRE_{sol}$ as function of the measured surface albedo. The colors represent the flights (days) during which the respective measurements were obtained. (Figure not included in the manuscript)*

this time period, the cloud field above the aircraft transitioned from overcast to cloud-free conditions and mostly consisted of cumulus clouds (Fig. 4). According to Mol et al. (2023), such transition periods with broken cumulus clouds are predestinated to cause $F_{sol}^{\downarrow}$ values exceeding the cloud-free $F_{sol}^{\downarrow}$ (see their Figs. 1 and 4).

[Figure]

*Figure 3: Time series of the measured and the cloud-free simulated $F_{sol}^{\downarrow}$ (upper panel), the measured and retrieved cloud-free albedo (middle panel) and the obtained $CRE_{sol}$ using the measured or cloud-free albedo (lower panel). (Figure not included in the manuscript)*

Broken cloud situations like this were not infrequently sampled during AFLUX over open ocean. Thus, these observations shape a mode peaking at positive $CRE_{\mathrm{sol}}$ values.

[Figure]

*Figure 4: Broken cumulus cloud situations on 08 April 2019 during the low-level leg between 11:15 and 11:35. (Figure not included in the manuscript)*

To avoid this partly misleading discussion, we omitted the term "cloud-free mode" in the revised manuscript. Instead we wrote: "**For AFLUX (Fig. 7d), mode 1 over open ocean shows a remarkably positive $CRE_{\mathrm{sol}}$ with a median of 20 W m$^{-2}$ for an equivalent LWP of less than 5 g m$^{-2}$. This solar warming effect is due to broken cumulus clouds, which often enhance $F_{\mathrm{sol}}^{\downarrow}$ compared to a cloud-free situation for several minutes by scattering additional solar radiation towards the surface (cloud enhancement, Mol et al., 2023). Broken clouds frequently occur during cold air outbreaks over open ocean, when the cold air advected over the warm ocean reduces the thermodynamic stability and leads to the formation of cloud streets (e.g., Brümmer, 1996[, https://doi.org/10.1007/BF00119014]). Thus, mode 1 combines cloud-free situations with broken cloud observations, the latter producing a low retrieved equivalent LWP that is indistinguishable from cloud-free conditions.**"

*Line 71: Awkward wording. Maybe "Fewer efforts have focused on CRE over open…"*

Yes, we agree that this wording could be improved. Actually, we wanted to state that "**Less attention has been paid to the CRE over open (ice-free) ocean…**" and changed it accordingly in the revised version of the manuscript.

*Line 105: It might be better to show Eq (1) using terms for both TIR and solar separately to make cross-referencing like this more clear.*

Since we introduced the symbols in the new equations 1–4 (see answer to your earlier comment on that), we are now able to use them here and rearrange the text as follows: "**During the low-level sections, the broadband irradiances $F_{\mathrm{sol}}^{\downarrow}$ and $F_{\mathrm{sol}}^{\uparrow}$ on the one hand, and $F_{\mathrm{TIR}}^{\downarrow}$ and $F_{\mathrm{TIR}}^{\uparrow}$ on the other hand were measured by pairs of upward- and downward-directed pyranometers (sensitive in the solar range between 0.2–3.6 µm) and pyrgeometers (sensitive in the TIR range between 4.5–42 µm) at a frequency of 20 Hz. [comment below] From the broadband irradiance measurements, $F_{\mathrm{net,sol,cld}}$ and**

$F_{\mathrm{net,TIR,cld}}$ (Eqs. 1 and 2), and the surface albedo (ratio of $F_{\mathrm{sol}}^{\uparrow}$ and $F_{\mathrm{sol}}^{\downarrow}$) in mostly cloudy conditions were derived."

*Line 105/111: This seems deceptive. What is the response time of the thermopiles? Are you really making independent samples at 20 Hz?*

*Line 107: While discarding tilted data is one approach, even at 5 deg biases can be large. Corrections are possible up to 10 deg (https://www.doi.org/10.2174/1874282301004010078). Did you consider this?*

Note that this answer refers to the previous two comments:

Unfortunately, we forgot to mention a couple of facts regarding the processing of our data, which we of course did in advance and which is described in detail in two other papers (Ehrlich et al., 2019; Mech et al., 2022).

It is correct that, due to the response time of the thermopiles (in the order of few seconds), the recorded 20 Hz data is not independent. We applied a deconvolution method to partly reconstruct the high frequency variability in the data and to correct the temporal shift of the time series induced by the sensor inertia (see Ehrlich and Wendisch, 2015, https://doi.org/10.5194/amt-8-3671-2015). This correction requires smoothing of instrument noise and, therefore, still does not provide independent 20 Hz data. Nevertheless, fluctuations can be resolved up to about 2 seconds, which is still below the response time of the thermopiles.

We also corrected the solar downward irradiance for aircraft attitude. Compared to the suggested reference (Long et al., 2010), we don't have separate measurements of the direct and diffuse components. Thus, we chose an intermediate way. We roughly divided the data set into sections with cloudy (dominated by diffuse radiation) and cloud-free (dominated by direct radiation) conditions and applied a correction to the latter based on cloud-free radiative transfer simulation of the direct and diffuse fractions. Since this is of course less accurate than the method of Long et al. (2010), we additionally discarded data with roll/pitch angles exceeding 5°.

Together with the two references (Ehrlich and Wendisch, 2015; Bannehr and Schwiesow, 1993), the following passage, which is added to the text, should make this clear: "**… and recorded at a frequency of 20 Hz. An inertia correction was applied, which enables to resolve fluctuations in the order of 2 s and to remove the inertia-induced time shift of the time series (Ehrlich and Wendisch, 2015). Furthermore, the impact of the aircraft attitude on $F_{\mathrm{sol}}^{\downarrow}$ was accounted for by a common correction method (Bannehr and Schwiesow, 1993). Because of remaining uncertainties in the estimation of the fraction of direct solar irradiance, the irradiance data for aircraft attitudes exceeding 5° in roll and pitch angle were discarded.**"

*Line 172: meidan -> median*

Changed.

*3.1: Here again, I think it would be useful consider Eq (1) as LW and SW separately and to distinguish more clearly in this paragraph the methodologies used for the two bands.*

Similar to the comment on line 105, we now use the symbols in the text. We started with: "**Section 2.1 describes the (mostly cloudy) measurements of $F_{\mathrm{net,sol,cld}}$ and $F_{\mathrm{net,TIR,cld}}$. To calculate both $CRE_{\mathrm{sol}}$ and $CRE_{\mathrm{TIR}}$ (Eqs. 3 and 4), $F_{\mathrm{net,sol,cf}}$ and $F_{\mathrm{net,TIR,cf}}$, need to be simulated.**"

To highlight, which input is used for the simulations of both $F_{\mathrm{net,sol,cf}}$ and $F_{\mathrm{net,TIR,cf}}$, we added "**For both spectral ranges…**". When it comes to the differences between the solar and the TIR ranges, we tried to separate more clearly, which input is needed for which range: "**In addition to these** (the common) **settings, $F_{\mathrm{net,TIR,cf}}$ was simulated using a surface emissivity of 0.99 … Instead of the surface emissivity, the simulation of $F_{\mathrm{net,sol,cf}}$ additionally requires the SZA and the definition of the local surface albedo … From the simulations, the direct/diffuse fraction of $F_{\mathrm{sol}}^{\downarrow}$ was obtained for cloud-free conditions.**"

*Figure 4: Maybe specify in the caption that this is simulated, not observed.*

Yes, that makes total sense to not confuse the reader. We added "**Simulations of the …**" at the beginning of the figure caption for both (a) and (b).

*Figure 8: So just to be clear, there were no clear-sky samples made during MOSAiC-ACA? I think it might be helpful to emphasize that point because to look at Figure 8 it appears as if you are reporting CRE > 25 Wm2 under clear skies.*

We agree that the two modes in the distributions of MOSAiC-ACA could cause confusion if they are compared to the cloud-free and cloudy modes found for the other campaigns.

In the original manuscript, we mentioned that the two-mode structure of cloudy and cloud-free mode is only present for AFLUX and ACLOUD: "**Independent of the underlying surface type, the distributions of $CRE_{\mathrm{TIR}}$ during AFLUX (Figs. 8a, 8d) and ACLOUD (Figs. 8b, 8e) reveal two distinct modes. Similar to the $CRE_{\mathrm{sol}}$, the mode located around 0 W m$^{-2}$ represents cloud-free conditions, while the second mode clearly indicates the warming effect of the clouds in the TIR range.**"

Later, when discussing MOSAiC-ACA, we explain that none of the modes represents actual cloud-free conditions (as we did before when discussing the solar CRE). We slightly adjusted the text to be more clear: "**Due to the lack of cloud-free observations, the mode with the smallest $CRE_{\mathrm{TIR}}$ represents the broken cloud conditions (around 25 W m$^{-2}$).**"

To also emphasize this fact when only looking at Fig. 8, we decided to add the following sentences to the figure caption: "**Note that, due to a lack of the corresponding observations, none of the modes in (c) represents actual cloud-free conditions. The thinnest clouds, however, revealed an equivalent LWP < 5 g m$^{-2}$ and did, thus, not contribute to the statistics shown in (f).**"

---

## Author Comment (AC2)

**Answers to comments of Review #2**

We gratefully thank the reviewers for the positive feedback on our submitted manuscript. We appreciate the time they took to extensively read and comment on the given manuscript. The constructive comments are very helpful for the improvement of the manuscript. Our replies to the referees' comments are structured as follows:
*Referee's comments in italic – line numbers according to initially submitted manuscript*
Authors' responses in roman – line numbers according to adjusted manuscript. **Citations from the initial and the adjusted manuscript are given in bold.**

*2.1 Major Comments*

*Consider changing the terminology of "solar" and "thermal-infrared" to shortwave and longwave (and later abbreviated as "SW" and "LW", even the spectral ranges of the broadband radiometers only partially cover the SW and LW) since the SW and LW are more broadly used by the CRE community.*

Although the terms "shortwave" and "longwave" are more widely used, we prefer to use terms that, in our point of view, more precisely define the measured quantity. Whether a wavelength is shortwave or longwave always depends on the perspective, these terms are somehow relative (Bohren and Clothiaux, 2006). For people working with microwaves, also the so-called "longwave" radiation has short wavelengths. Thus, we use the term "solar" instead of "shortwave", which clearly indicates that almost the entire radiation emitted in this wavelength range originates from the Sun. In contrast to Bohren and Clothiaux (2006), we use the term "thermal-infrared" instead of "terrestrial" or "longwave" to indicate that this wavelength range, which is part of the infrared, mostly consists of thermal radiation emitted by objects with temperatures typical for the Earth's atmosphere. The term "terrestrial" might be misleading, since this radiation is not only emitted by the Earth, but also by atmospheric objects.

In the end, the usage of the terms is a rather philosophic question without true and false answers. It is, however, important that the wavelength ranges are specified, which we do in our manuscript.

We added a footnote to the manuscript: "**The terms "solar" and "thermal-infrared" are often referred to as "shortwave" and "longwave". However, since the latter terms might be relative, we use the former terms throughout this manuscript (Bohren and Clothiaux, 2006, page 22f.).**"

However, we abbreviate the terms in the remainder by adding the subscripts "sol", "TIR" (and "tot") to the variables.

*L80: "However, both results included … observations." I didn't quite get what are the limitations of others studies. Please clarify.*

Here, we wanted to mention that the similarity of the CRE found by the satellite observations (Kay and L'Ecuyer, 2013) and the study of Ebell et al. (2020) at Ny-Ålesund could result from the charactieristic surface types of these observations. In contrast to the other studies, which performed measurements only over bright sea ice or snow, also snow-free surfaces were observed by Kay and L'Ecuyer (2013) and Ebell et al. (2020). To clarify this, we changed the sentence to: "**This similarity likely results from the**

**observation of snow-covered and snow-free surfaces in both studies, while most other aforementioned studies only investigated snow-covered surfaces.**"

*For Figure 1, considering changing the color of red (or orange) to another color (green maybe?) and make the lines for low level sections slightly thinner (or add a little bit transparency in color) so the flight tracks can be better distinguished.*

To increase the contrast between all flight tracks and the low-level sections, we changed the light red color indicating all flight tracks to a brighter orange. The dark red lines for the low-level sections were thinned, as suggested. Additionally, we added LYR (Longyearbyen) as campaign base to the map. See the revised figure below. The first sentence of the figure caption now reads: "**Flight tracks (orange) and low-level sections (dark red) performed during (a) AFLUX, (b) ACLOUD, and (c) MOSAiC-ACA based at Longyearbyen (LYR).**"

[Figure]

*For Figure 2, consider changing the dashed line (or dotted line) to solid line, as dashed lines and dotted lines are difficult to distinguish. Also, the temperature variation of profile near surface is almost invisible, can you experiment with log y axis to see whether the temperature variation near surface stands out more (e.g., temperature inversion)?*

We changed the dashed lines to solid lines. However, rather than changing to a log y axis, we removed the values below 90 m from the profiles of both temperature and absolute humidity. Since only a small number of aircraft ascents/descents reached such low altitudes, the sparse sampling statistics artificially shape the temperature and absolute humidity profiles there. Thus, the interpretation of these values is difficult and not valuable. Additionally, the exact knowledge of the surface temperature is not important, because we assume the surface temperature and, consequently, the surface emission to not change between cloudy and cloud-free conditions. The revised Figure 2 is similar to Fig. 3 shown in these replies, except that the relative humidity is not shown in the manuscript.

Because of the exclusion of the lowest altitudes, we needed to change the sentence "**Although the temperature at the open ocean surface was close to the freezing point during all campaigns, it strongly decreased within the lowermost 80 m during AFLUX.**" to "**The near-surface temperature over open ocean was close to the freezing point during ACLOUD and MOSAiC-ACA, and below -10 °C during AFLUX.**"

*L181-185: Which do you think is the more plausible cause for the much more frequent thin clouds occurrence observed during MOSAiC-ACA? Limitation in sampling statistics (e.g., with more data we will see similar distribution like ACLOUD) or the cloud type associated with Arctic season (e.g., even with more data we will still see predominant thin clouds)?*

We think that especially the low sampling statistics is the reason for the distribution of the equivalent LWP over sea ice during MOSAiC-ACA. The respective data consists of three sets of maximum two-minute samples, two of which were performed within 10 minutes, the third one approximately 45 minutes later. Within this time, we assume the cloud conditions to not have changed significantly, leading to this narrow distribution. Statistically, this is not representative for the sea ice conditions during that season. For a larger sample, we would at least expect a broader distribution and a larger median LWP. Whether this LWP really approaches the ACLOUD distribution is speculative. Climatologies of Arctic cloud properties are sparse. According to Wang and Key (2005, https://doi.org/10.1029/2004JD005720, their Fig. 5d), the mean cloud optical thickness in the Arctic ocean region in September is only slightly lower than in March. In the North Pole regions, the values are similar in September and March/April, but significantly lower than in summer. We added to the text: "**These observations are statistically not representative and very likely don't reflect typical conditions present over sea ice during this season.**"

*Figure 5 is very interesting. I would like to see more explanation about why the albedo change in such way (e.g., the "dip" of albedo at SZA of 75° when transitioning from clear-sky to optically very thin clouds, there must be some counteracting factors) rather than the descriptions of how the albedo change along LWP.*

As explained in subsection 3.2.1, two effects (geometry effect and spectral weighting effect) contribute to the change of the broadband surface albedo, i. a., depending on SZA and cloudiness.

It is shown that the geometry effect dominates over the spectral weighting effect for open ocean, which leads to an albedo decrease with increasing cloudiness (LWP). This is already explained in the original text: "**The broadband open ocean albedo (Fig. 5a) decreases with increasing LWP, which indicates that the geometry effect dominates over the spectral weighting effect. This is due to the relatively low spectral differences of the spectral open ocean albedo (Fig. 4a).**" The surface albedo differences between the different SZAs for cloud-free conditions and optically thin clouds are also explained: "**Similar to the spectral albedo, … the albedo in cloud-free conditions increases for increasing SZA.**" The reason for a higher albedo and higher SZA is explained in Sect. 3.2.1: "**This difference is due to the enhanced specular reflection at the air–water interface for larger incident angles (i. e., SZA), according to Fresnel's equations.**"

For sea ice, the spectral weighting effect mostly surpasses the counteracting geometry effect, leading to an albedo increase with increasing cloudiness (for details, we referred to Stapf et al. (2020) who already did similar analysis for the sea ice albedo): "**Thus, the spectral weighting effect becomes more dominant and leads to an increase of the broadband albedo with increasing LWP (Fig. 5b).**" We agree that the reason of the "dip" occurring for a SZA of 75° is missing. We add the following sentence: "**This feature arises from the geometry effect surpassing the spectral weighting effect for optically thin clouds when the Sun is low enough.**"

*Consider changing the color of the markers (crosses and dots, also in the legend) in Figure 7 and 9 as they are hardly distinguishable between the numbers and add descriptions in the figure caption.*

Instead of changing the color of the markers, we changed the color of the numbers to grey (see below). Hopefully, markers and numbers are now better distinguishable. In the figure captions, we changed the last sentence to "**The symbols represent the median of $CRE_{sol}$ and the measured surface albedo** (Fig. 7) / $CRE_{sol}$ **and** $CRE_{TIR}$ (Fig. 9) **over the surface types given in the legend in (a) and only considering cloudy observations (equivalent LWP > 5 g m$^{-2}$).**" to better refer to the markers. Additionally, we added a sentence explaining the numbers of the modes: "**The numbered modes represent (1) cloud-free open ocean, (2) cloud-free sea ice, (3) cloudy open ocean, (4) cloudy sea ice, (5) thin/broken clouds, and (6) cloudy MIZ conditions.**" (Fig. 7) and "**The numbered modes represent (1) cloud-free, (2) cloudy open ocean, (3) cloudy sea ice, (4) thin/broken clouds, and (5.1/5.2) cloudy MIZ conditions.**" (Fig. 9).

[Figure]

[Figure]

*Figure 1: Same as Fig. 7 in the manuscript, but for $CRE_{sol}$ normalized with the cosine of the SZA on the y-axis. (Figure not included in the manuscript)*

*From L316-322, the author argues the SZA causes the different distributions in CRE among different campaigns. I have an idea of normalizing the CRE with the cosine of SZA (it should not be difficult to do). If SZA is the culprit, the distribution difference should disappear once the CRE is normalized.*

We also normalized the $CRE_{sol}$ with the cosine of the SZA during our analysis. The result is shown in Fig. 1, which indicates that the modes of all campaigns are similar when normalized $CRE_{sol}$ values are analyzed. Thus, the SZA is clearly responsible for the $CRE_{sol}$ differences between the campaigns. Merging the observations of all campaigns (accounting for the different number of data points such that all campaigns are weighted equally) leads to a distribution with four clearly separated modes (Fig. 2). In the manuscript, we decided to keep the original plot with the non-normalized $CRE_{sol}$ as this provides an absolute value of $CRE_{sol}$, which finally contributes to the surface energy budget. We modified the respective sentence in the text: "**Although the lower surface albedo contributed to the lower $CRE_{sol}$ during ACLOUD, a normalization of $CRE_{sol}$ with the cosine of the SZA (not shown) reveals that the major contribution to the $CRE_{sol}$ differences between the two campaigns resulted from the different solar illumination as a consequence of the clearly distinct SZA ranges (Table 1).**"

*I quite like the places where you brought up "broken clouds" seen during the MOSAiC-ACA campaign, which have caught my attention in wondering how much 3D cloud radiative effects are there in the Arctic. Technically, the 3D effects should be predominant in the Arctic when broken clouds present as the surface is bright and sun is low (more scattering events). Even though I understand the 3D effects are not the focus of this paper, I would recommend adding some brief discussion about it could potentially favor the radiation closure development in the Arctic.*

[Figure]

*Figure 2: Same as Fig. 5, but the observations of all campaigns are merged such that all campaigns have equal weighting. (Figure not included in the manuscript)*

We agree that quantifying the 3D effects is not the scope of our study. However, we provide some brief discussion about the topic here: Of course, if we would only be interested in $F_{sol}^{\downarrow}$, multiple scattering occurring between surface and cloud will be stronger over bright sea ice than over dark open ocean surfaces. When interpreting the 3D cloud radiative effects on $CRE_{sol}$, the aforementioned effect is counteracted by the surface albedo. Over bright sea ice, an enhanced $F_{sol}^{\downarrow}$ is counterbalanced by an enhanced reflection at the surface. Thus, the magnitude of $CRE_{sol}$ over sea ice is always lower than over open ocean. For this reason, 3D effect will appear more dominant over open ocean and are less imprinted in $CRE_{sol}$ over sea ice. Besides that, cloud types might be different over the different surfaces. We observed broken clouds and the associated effects only over open ocean, mostly during cold air outbreaks (CAOs). When the cold air is advected over the relatively warm open ocean, the stability is significantly reduced, leading to roll convection and forming cloud streets. These cloud streets consist of broken clouds with gaps in between. We discuss the broken clouds for the first time in Sect. 4.1 (Solar CRE) for AFLUX. There we add: "**This solar warming effect is due to broken cumulus clouds, which often enhance $F_{sol}^{\downarrow}$ compared to a cloud-free situation for several minutes by scattering additional solar radiation towards the surface (cloud enhancement, Mol et al., 2023**[, https://doi.org/10.1029/2022JD037894]**). Broken clouds frequently occur during cold air outbreaks over open ocean, when the cold air advected over the warm ocean reduces the thermodynamic stability and leads to the formation of cloud streets (e.g., Brümmer, 1996**[, https://doi.org/10.1007/BF00119014]**)."**

*Consider adding some thoughts about lesson learned from the three Arctic aircraft campaigns and improvement one can make for future aircraft campaigns in order to progress cloud radiation science in the Arctic (e.g. ARCSIX) in the conclusion.*

We added: "**The results might be biased by the flight strategy and the spatial and temporal selection of low-level sections to satisfy different campaign goals. Furthermore, not all synoptic conditions could be captured due to weather-caused flight limitations. To overcome these limitations and to improve the statistics of the surface CRE, extensive low-level sections are necessary regardless of the weather conditions.**" and later "**The validation of satellite CRE retrievals with airborne measurements might be the key for long-term observations of the CRE over open ocean. This study and the published datasets of the CRE in the Fram Strait (Stapf et al. 2021c, Becker et al., 2023) could provide a basis for such investigations and for further research of cloud-related processes and feedback mechanisms in numerical models.**"

*2.2 Minor Comments*

*L1: <during airborne> to <from>; <three> to <three airborne>*

Thanks for the suggestion, we changed it accordingly.

*L2: suggest to added <– AFLUX (2019 March to April), ACLOUD (2017 May to June), and MOSAiC-ACA (2020 August to September)> after <Svalbard>*

We added this information: "**…: AFLUX (March/April 2019), ACLOUD (May/June 2017), and MOSAiC-ACA (August/September 2020)**"

*L3: <of the surface> to <at the surface>*

Ok.

*L8: <component> to <components>*

Changed.

*L8: <and in combination> to <as well as combined for the study of total CRE.>*

Changed accordingly.

*L90: <airborne measurements of … were performed during three seasonally distinct campaigns in the vicinity of Svalbard> to <three airborne campaigns were deployed to collect measurements of cloud, surface, and thermodynamic properties during different seasons near Svalbard.>*

Changed accordingly.

*L97: <performed> to <deployed>*

Ok.

*L108: add brief description of why the observations are discarded, something like <due to the contamination of radiation signals from …> after <discarded>*

Due to another reviewer's comment, the entire sentence is changed to: "**Because of remaining uncertainties in the estimation of the fraction of direct solar irradiance, the irradiance data for aircraft attitudes exceeding 5° in roll and pitch angle were discarded.**"

*L121: <the cloud boundaries> to <cloud boundaries>*

Changed.

*L134: <fice> was not clarified (or I missed it?)*

You are correct, we forgot to introduce the variable $f_{ice}$, representing the sea ice concentration. We included it in the sentence explaining the measurement of $f_{ice}$ with the fish-eye camera: "**Additionally, the sea ice concentration $f_{ice}$ was derived from measurements of a three-channel digital camera equipped with a 180° fish-eye lens (sampling frequency: one image every 6 seconds).**"

*L151: worth adding RH profiles*

We decided against the depiction of the RH profiles for several reasons: First, another panel would reduce the lucidity of the plot. Second, rather than the RH, the AH representing the total amount of water vapor is directly related to the radiative emission by water vapor. Third, the information content delivered by the RH profiles is minimal. We additionally show the RH profiles here (Fig. 3). The important points discussed in the text become obvious: In general, the RH is lower during ACLOUD compered to MOSAiC-ACA, which causes the larger AH during the latter campaign. Additionally, the AH differences between sea ice and open ocean for ACLOUD and MOSAiC-ACA (at least below 1500 m) can be explained by the differences in RH.

*L172: <meidan> to <median>*

Changed.

*L203:  do you mean <solar geometry>? Just want to confirm.*

[Figure]

*Figure 3: Same as Fig. 2 in the manuscript, except the additional panel showing the mean profiles of relative humidity (RH). OO – open ocean, SI – sea ice. (Figure not included in the manuscript)*

Partly yes. Beside solar geometry (solar zenith and azimuth angles), we additionally mean by that whether the incident solar radiation is rather direct (as for cloud-free conditions) or diffuse (as in cloudy conditions). We changed it to "**amount of direct and diffuse solar radiation**"

*L230: why? Due to rough ocean surface from high wind speed, thus less specular reflection?*

Exactly, we add the following phrase to the beginning of the sentence: "**Due to a reduction of specular reflection on a roughened surface, …**"

*L247: where does <sea ice concentration> come from? Estimation from aircraft camera imagery? Any reference for figure for this linear combination?*

Please see the answer to the comment on L134. The sea ice concentration $f_{ice}$ was derived from camera images. To clarify, we replaced "**the sea ice concentration**" by "**… $f_{ice}$ obtained from the fish-eye camera (Becker et al., 2022)**" and gave a reference.

We noticed that "sea ice concentration" and "$f_{ice}$" were used inconsistently. We thus replaced all but the first occurrence of "sea ice concentration" by "$f_{ice}$".

*L251: <amounts to> to <converges at>*

Ok.

*L258: see my earlier comments, why a decrease is observed for SZA of 75 but not 60 under clear-sky?*

See my earlier answer on that comment. ;)

*Figure 6: consider adding "retrieved" and "observed" to the y axis labels for (d) to (f), as well as for the legend labels in (b)*

We do not see the benefit of changing "measured" to "observed" and kept it. In contrast, to make clear that the shown "**cloud-free**" albedo was retrieved, we decided to combine both terms to "**retrieved cloud-free**" conditions in both figures. We also changed it in the text, where appropriate.

*L284: <their mode structures> to <the mode structures over the parameter spaces of surface albedo and CRE>*

With your suggestion and keeping the following sentence, this information would be doubled. Thus, we decided to directly combine the sentence with the following one: "**… the frequency distributions of $CRE_{sol}$ as a function of the strongly influential measured surface albedo, which are shown in Fig. 7.**"

*L287: add <cloud-induced> before <surface albedo change>*

Added.

*Figure 7: last sentence is unclear, please clarify which symbols are which. Consider referencing back to the text or adding an example to explain. Consider change the color of dot and cross markers to distinguish them from numbers.*

See my answer to your major comment on that. The symbols are clearly attributed to the different surface types using the legend in (a). We clarified that by changing the last sentence as specified in the earlier answer.

*L290: since you are showing broadband irradiance, suggest change <solar spectral range> to <solar range>.*

The terms were used inconsistently throughout the manuscript. We decided to omit the term "**spectral range**" entirely and replaced it by simply "**range**".

*L291: <Figs. 7d, 7f> should be <Figs. 7c, 7f>*

Thank you, we changed it.

*L292: can the mode change/shift due to 3D cloud radiative effects (from thin or broken clouds)?*

Please see my answer to your major comment on the broken cloud effect.

*L300: consider providing the actual values of solar CRE of mode 4 in Fig. 7e*

The actual value of mode 4 in Fig. 7e is given later in the text (**-65 W m$^{-2}$**). Thus, we think that the information "**almost doubled**" in combination with the of -33 W m$^{-2}$ for mode 4 in Fig. 7b is sufficient here.

*L301: <reduction> can be unclear about which way CRE goes, whether more cooling (values become more negative) or more warming (values become more positive). Consider <mitigation>.*

We agree. Thus, we replaced this and similar occurrences by the terms "**weakening**" and "**strengthening**" of the solar cooling effect. Larger/lower solar cooling is changed to "**stronger**" and "**weaker**" (or similar) solar cooling effect. Hopefully, the new terms will be associated less with numbers than the old ones.

*L303: <was increased by 29 Wm-2> to <imposed an artifact of 29 Wm-2 cooling due to the neglect of cloud-induced surface albedo change>*

Ok.

*L304: <slight> to <negligible>*

Ok.

*L309: see my earlier comments, consider mentioning the 3D cloud radiative effects*

See my earlier answer.

*L323:  to *

Ok.

*L341: the cross marker in Figure 7 seems unexplained.*

Actually, none of the markers is unexplained. The legend in Fig. 7a explains the markers. As mentioned earlier, we refer to this legend in the figure caption, such that, hopefully, it becomes clear what is meant by the markers.

*L456: reference shown up as <?>*

*L470: please fix <?, …>*

These two comments have a joint answer.

This "?" is the placeholder for a reference of a data set, which has been submitted recently during the review process. As the reference for this data set is now available, the question marks are replaced by the actual reference (Becker et al., 2023, https://doi.pangaea.de/10.1594/PANGAEA.95775).